# Neutrophil activation and clonal CAR-T re-expansion underpinning cytokine release syndrome during ciltacabtagene autoleucel therapy in multiple myeloma

Shuangshuang Yang [1,4], Jie Xu [1,4], Yuting Dai [1,4], Shiwei Jin [1], Yan Sun [1], Jianfeng Li [1], Chenglin Liu [1], Xiaolin Ma [1], Zhu Chen[1], Lijuan Chen[2], Jian Hou [3], Jian-Qing Mi[1] ✉ & Sai-Juan Chen [1] ✉

Cytokine release syndrome (CRS) is the most common complication of chimeric antigen receptor redirected T cells (CAR-T) therapy. CAR-T toxicity management has been greatly improved, but CRS remains a prime safety concern. Here we follow serum cytokine levels and circulating immune cell transcriptomes longitudinally in 26 relapsed/refractory multiple myeloma patients receiving the CAR-T product, ciltacabtagene autoleucel, to understand the immunological kinetics of CRS. We find that although T lymphocytes and monocytes/macrophages are the major overall cytokine source in manifest CRS, neutrophil activation peaks earlier, before the onset of severe symptoms. Intracellularly, signaling activation dominated by JAK/STAT pathway occurred prior to cytokine cascade and displayed regular kinetic changes. CRS severity is accurately described and potentially predicted by temporal cytokine secretion signatures. Notably, CAR-T re-expansion is found in three patients, including a fatal case characterized by somatic *TET2*-mutation, clonal expanded cytotoxic CAR-T, broadened cytokine profiles and irreversible hepatic toxicity. Together, our findings show that a latent phase with distinct immunological changes precedes manifest CRS, providing an optimal window and potential targets for CRS therapeutic intervention and that CAR-T re-expansion warrants close clinical attention and laboratory investigation to mitigate the lethal risk.

Over a decade, chimeric antigen receptor modified T cells (CAR-T) therapy emerges to be an attractive approach against hematological malignancy, marking a cancer treatment paradigm shift. The overall therapeutic effectiveness of CAR-T is quite encouraging, but its toxicity varies individually from clinically manageable to life-threatening. Cytokine release syndrome (CRS) is the most common adverse effect after CAR-T infusion. Severe CRS (Grade ≥ 3) is reported in nearly 46% of B-cell acute lymphoblastic leukemia (ALL)[1] and 13% of

[1]Shanghai Institute of Hematology, State Key Laboratory of Medical Genomics, National Research Center for Translational Medicine at Shanghai, Ruijin Hospital Affiliated to Shanghai Jiao Tong University School of Medicine, Shanghai 200025, China. [2]Department of Hematology, First affiliated Hospital of Nanjing Medical University, Jiangsu Province Hospital, Nanjing 210029, China. [3]Department of Hematology, Ren Ji Hospital affiliated to Shanghai Jiao Tong University School of Medicine, Shanghai 200127, China. [4]These authors contributed equally: Shuangshuang Yang, Jie Xu, Yuting Dai. ✉e-mail: jianqingmi@shsmu.edu.cn; sjchen@stn.sh.cn

B-cell lymphoma[2] patients receiving anti-CD19 CAR-T therapy, and 41% of multiple myeloma (MM)[3] patients infused with anti-BCMA CAR-T.

CRS is theoretically an indispensable inflammatory response during CAR-T therapy. Appropriate CRS is thought to be helpful to mitigate cancerous cells. However, excessive CRS can result in vital organ injury, and often put patients at risk. Hence, an in-depth understanding of the CRS mechanism of action is warranted to properly manage severe immune response. Serological tests show that CRS occurs with a robust elevation of diverse cytokines[4], of which, IL-6, IL-1β and GM-CSF, mediated by the monocyte/macrophage signalings, play crucial roles in cytokine storm development[5–7]. However, the comprehensive chronological pathophysiologic landscape of CRS upon CAR-T therapy hasn't yet been fully elucidated.

Ciltacabtagene autoleucel (Cilta-cel, previously known as LCAR-B38M) is a biepitopic CAR-T product targeting BCMA antigen, and has achieved remarkable success in the treatment of relapsed/refractory (r/r) MM patients in China[3,8,9] and United States of America (US)[10]. In light of it, Cilta-cel successively obtained approval from the US FDA, the European Medicines Agency, and the regulatory agency of Japan[11] recently.

In the present work, peripheral blood (PB) samples are collected from r/r MM patients who received Cilta-cel. Longitudinal cytokine profiling and gene transcriptomes are performed to characterize the dynamics of acute inflammatory events in response to CAR-T. We reveal a kinetic change in the axis of signaling activation-cytokine release-clinical manifestation, showing a stepwise evolution of CAR-T-related acute inflammation. We also point out that multiple factors may be able to collectively trigger CAR-T re-expansion, and subsequently prompt a toxic reaction that is probably severer than tumor-alone-driven CRS.

This study provides a solid rationale for a timely and precise application of targeted drugs, and proposes a notion that CAR-T re-proliferation requires more close monitoring and effective medical management to alleviate the lethal severity.

## Results

### Patient characteristics, clinical response and study design

The samples investigated were obtained from 26 Cilta-cel-exposed r/r MM patients, of whom, 16 cases were from phase I Legend-2 trial as we reported previously[3], and 10 from phase II CARTIFAN-1 trial[9]. Patients' demographic data are summarized in Table 1. Statistically, there were no significant differences of age, gender, MM subtype and other baseline characteristics between the two subpopulations from the above-mentioned separate trials (Table 1).

As of the manuscript preparation, the overall response rate of the 26 cases was 88.5%, with 80.8% obtaining complete responses (CR). The progression-free survival (PFS) rates were 39.2% at 3 year and 26.1% at 5 year; the overall survival (OS) rates were 53.1% at 3 year and 45.5% at 5 year (Supplementary Fig. 1a-b).

PB specimens were collected at the given time points. Serum was separated from PB for cytokine profiling. Purified mononuclear cells of PB (PBMCs) were assigned to three experimental cohorts as shown in Fig. 1a and Supplementary Data 1.

### Characterization of CRS

Within 30 days post Cilta-cel administration, CRS was observed in all patients at a median onset time of 6 days (range, 1-10 days). Fever signaled as an earliest event in 65.4% cases, appearing as early as 6 days at average. In the other 34.6% patients, hypotension, hypoxemia or organ malfunction presented earlier than fever in CRS outbreak. 34.6% patients had Grade 1-2 CRS and 65.4% suffered Grade 3 or worse (Fig. 1b). In accordance with clinical symptom, CRS grading was adjusted and the change of CRS grading aligned with the kinetic amount of circulating Cilta-cel (Fig. 1b-c). CD8:CD4 ratio of CAR-T increasingly elevated as CRS progressed, reaching the peak at day 16-20 (Fig. 1d). Based on the two typical T cell immunophenotyping markers CCR7 and CD45RA[12], cytotoxic CAR-T in CRS was enriched for effector memory (CCR7⁻ CD45RA⁻) and effector memory T cells re-expressing CD45RA (CCR7⁻ CD45RA⁺), particularly accumulated at day 10-12 (Fig. 1e).

To examine whether the baseline clinical and laboratory characteristics had an impact on CRS, we compared several key parameters between severe (grade > 2) and mild CRS (grade ≤ 2) groups. The patients subjected to combination lymphodepletion (Cyclophosphamide + Fludarabine) experienced more severe CRS than those receiving Cyclophosphamide alone (15/18 versus 2/8, $p = 0.008$), which was consistent with the observation previously stated[13]. At baseline, endogenous lymphocyte count was significantly lower in the severe CRS group than that in the mild group ($p = 0.016$). However, there was no statistical difference in the neutrophils ($p = 0.545$) and monocyte ($p = 0.199$) count between mild and severe CRS groups. With respect to the CAR-T, the mean frequencies of CD8⁺ and CD4⁺ CAR-T in the mild CRS group were respectively 59% (0.46 × 10⁶/kg) and

## Table 1 | Baseline characteristics

| | Total | Phase I | Phase II | p value |
|---|---|---|---|---|
| Case, n | 26 | 16 | 10 | |
| Median age (range) | 56.5 (35~73) | 55.5 (35~73) | 60 (52-64) | 0.133† |
| Sex, n (%) | | | | 1.000† |
| Male | 17 (65.4) | 10 (62.5) | 7 (70.0) | |
| Female | 9 (34.6) | 6 (37.5) | 3 (30.0) | |
| Type of multiple myeloma, n (%) | | | | 1.000† |
| IgG | 13 (50.0) | 8 (50.0) | 5 (50.0) | |
| IgA | 8 (30.8) | 5 (31.3) | 3 (30.0) | |
| IgD | 1 (3.8) | 1 (6.3) | 0 (0.0) | |
| Light chain | 4 (15.4) | 2 (12.5) | 2 (20.0) | |
| Kappa | 0 (0.0) | 0 (0.0) | 0 (0.0) | |
| Lambda | 4 (15.4) | 2 (12.5) | 2 (20.0) | |
| Nonsecretory multiple myeloma | 0 (0.0) | 0 (0.0) | 0 (0.0) | |
| Extramedullary lesion, n (%) | 4 (15.4) | 4 (25.0) | 0 (0.0) | 0.136† |
| Median number of prior therapy Lines (range) | 4 (3-11) | 4 (3-11) | 4 (3-5) | 0.358† |
| Autologous stem cell transplantation, n (%) | 11 (42.3) | 8 (50.0) | 3 (30.0) | 0.428† |
| Bone lesions, n (%) | 21 (80.8) | 12 (75.0) | 9 (90.0) | 0.617† |
| High-risk cytogenetic abnormalitiesᵃ, n (%) | 10 (40.0) | 6 (40.0) | 4 (40.0) | 1.000† |
| Prior therapies, n (%) | | | | |
| Proteasome inhibitor | 24 (92.3) | | | 0.749† |
| Bortezomib | 23 (88.5) | 13 (81.3) | 10 (100.0) | |
| Carfilzomib | 2 (7.7) | 2 (12.5) | 0 (0.0) | |
| Ixazomib | 2 (7.7) | 1 (6.3) | 1 (10.0) | |
| Immunomodulatory drugs | 23 (88.5) | | | 1.000† |
| Lenalidomide | 19 (73.1) | 10 (62.5) | 9 (90.0) | |
| Thalidomid | 12 (46.2) | 6 (37.5) | 6 (60.0) | |
| Pomalidomide | 1 (3.8) | 1 (6.3) | 0 (0.0) | |
| Median CAR-T infused dose (×10⁶/kg) (range) | 0.595 (0.28-1.52) | 0.575 (0.28-1.52) | 0.63 (0.42-0.79) | 0.336† |

†p value is calculated by t-test.
‡p value is calculated by Fisher's exact test. All tests are two-sided.
ᵃHigh-risk cytogenetic abnormalities are defined by the presence of the following abnormalities: del(17p), t(4;14), or t(14;16). One patient's cytogenetic information is not available.

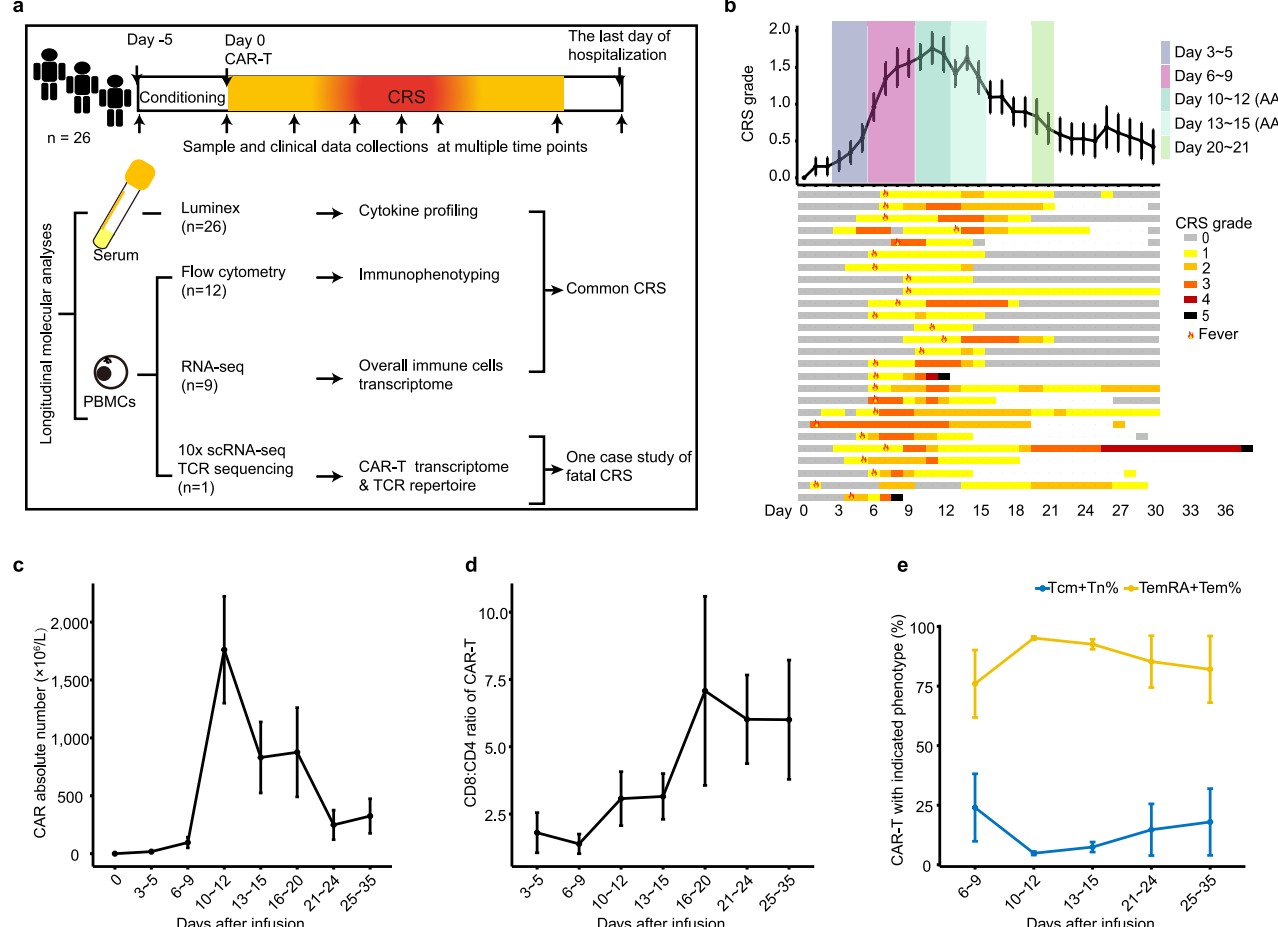

**Fig. 1 | Overview of the study design and dynamic changes of CAR-T kinetics. a** Overview of the study design including sample and clinical data collections, cytokine profiling, CAR-T immunophenotyping, transcriptome sequencing and TCR repertoire analyses. **b** The graph (top panel) shows the mean value of CRS grades in all the 26 patients on days after Cilta-cel infusion (mean ± SEM). The swimmer plot (bottom panel) shows the kinetics of the CRS severity in each patient within one month post infusions. One patient suffering lethal toxicity is shown a prolonged observation. Each row represents one patient. Colors on the bars indicate the grades of CRS on different days. The sign of 'flame' marks the onset of fever related to CRS. AA1/AA2 denote two time points at acute aggravation phase. **c** The proliferation of Cilta-cel in the peripheral blood ($n = 12$) shown by the absolute number (mean ± SEM). **d** The chart shows the CD8:CD4 ratio of CAR-T ($n = 9$). Data is presented as mean ± SEM. **e** The chart presents the immunophenotypic signature of CAR-T in the peripheral blood ($n = 7$). Data is shown as mean ± SEM. Tcm: central memory T cell (CCR7+ CD45RA−); Tn: naive T cells (CCR7+ CD45RA+); TemRA: effector memory RA+ T cells (CCR7− CD45RA+); Tem: effector memory (CCR7− CD45RA−) T cells. Source data are provided as a Source Data file.

41% ($0.25 \times 10^6$/kg), and those in the severe group were respectively 62% ($0.43 \times 10^6$/kg) and 38% ($0.32 \times 10^6$/kg). Neither CD8+ nor CD4+ reached proportionally and numerically significant differences between the two CRS groups. The killing capacity of CAR-T was evaluated by the apoptotic proportion of BCMA-expressing tumor in co-culture. Though phase I and phase II adopted different effector-to-target (E:T) ratios in vitro, there was no evident tumor apoptotic disparity between severe and mild CRS cohorts.

In addition, the individual therapeutic response to CRS in terms of CRS symptoms, grades, management, and outcome was summarized in Supplementary Data 2. We found no significant difference in clinical outcome between the patients with severe and mild CRS, suggesting clinical prognosis did not correlate with CRS grade.

**Dynamic changes of cytokine profile**
To gain insight into the development of CRS, patients' clinical presentation separated the entire 21-day inflammatory reaction into 5 slots, including baseline (before infusion), latent (day 3-5), fever (day 6-9), acute aggravation (day 10-15, AA), and resolving (day 20-21) periods. Since a robust cytokine secretion at the AA period, we collected individual samples two times (Day 10-12 and Day 13-15), denoted as AA1 and AA2, respectively to gain adequate information. Based on these time slots, serological data were accordingly assessed to draw an overall landscape of cytokine profile.

As compared with the baseline, the inflammatory molecules substantially up-regulated at the late fever period, reaching a high level at the acute aggravation period (Fig. 2a). Several well-studied cytokines, such as IL-6, Granzyme B, IL-10, G-CSF and CXCL10, rocketed up by 50-100 folds at the peak, indicating a highly active inflammatory response at the periods spanning from day 6 to 15 (Fig. 2b). By the means of unsupervised clustering, we identified five cytokine clusters (CCs), among which, CC3, CC4, and CC5 displayed a positive correlation with mean CRS grade, whereas CC2 exhibited a negative correlation (Fig. 2c). Notably, among the cytokines positively correlated with CRS severity, sIL-2Rα, sTNF RII (both belonging to CC4), and Granzyme B (belonging to CC5) exhibited the strongest correlations (Fig.2a, d Supplementary Data 3). To interpret the correlation of cytokine expression and CRS level, we adopted the computational methodology CONNECTOR[14] and partitioned the 26 patients into two groups: Group1 ($n = 8$) and Group2 ($n = 18$) (Fig. 3a-b). Further analysis of the mean log₂FC of cytokines revealed a distinct cytokine kinetic mode of Group1 from that of Group2 (Fig. 3c-d). The patients in Group1 displayed a higher peak level of CC3, CC4, and CC5 cytokines, as well as the whole, as compared with Group2. Based on this grouping, we can

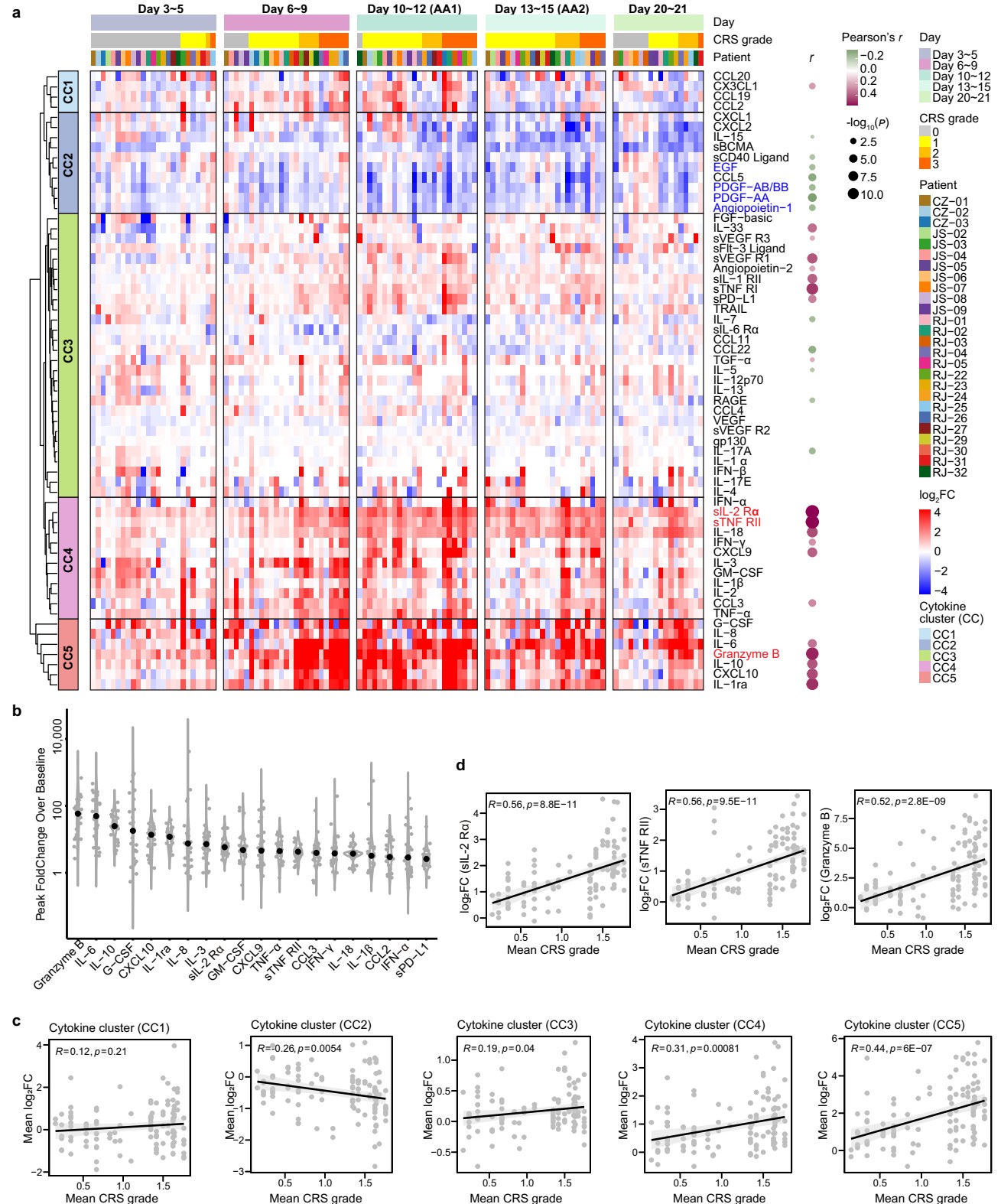

**Fig. 2 | Dynamic changes of cytokines. a** The heat map exhibits dynamic changes of 61 cytokines (*Y* axis) in 26 patients (color bars on *X* axis) at different time slots (*X* axis) after CAR-T therapy. The cytokine level is defined as the fold change (FC) of a certain cytokine value relative to its baseline, and transformed in the format of $\log_2$FC. The vertically arranged dots at the right side illustrate the correlation between the mean CRS grade and the cytokine level, of which the top three cytokines (sIL-2Rα, sTNF RII and Granzyme B) with the highest correlation coefficients are shown in details in (**d**). **b** The median peak FC of each cytokine ranked at the top 20 is shown in the plot. The peak FC for each evaluable case is shown as a grey dot, and the median FC value is shown in black. **c-d**, Scattered diagrams show the correlation between the mean CRS grade and the mean $\log_2$FC changes of cytokines (**c**), as well as the correlation between the mean CRS grade and the cytokine level (**d**). *p* values in **a**, **c** and **d** are calculated using two-sided Pearson's *r* correlation. Source data are provided as a Source Data file.

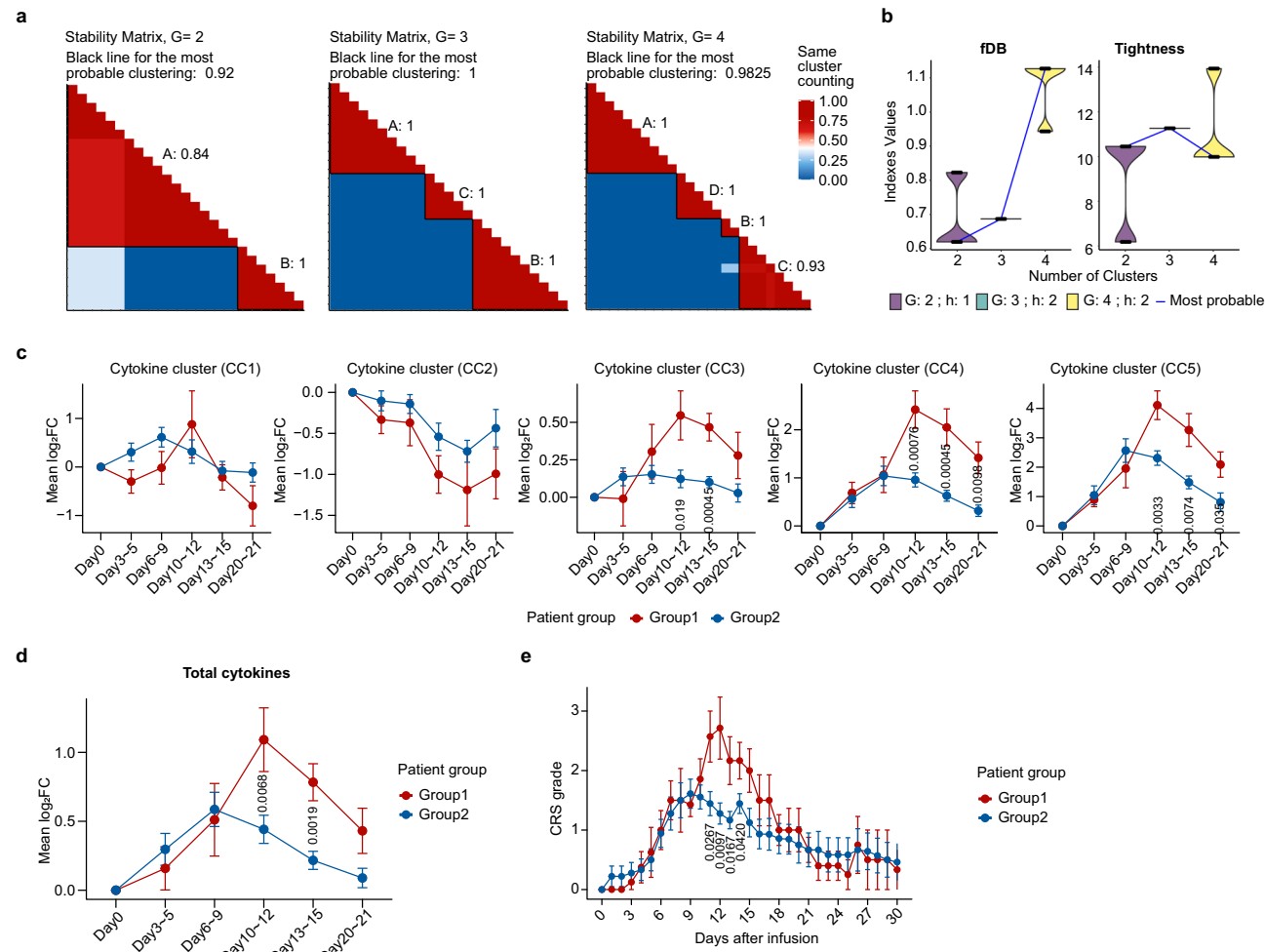

**Fig. 3 | Dynamic correlation between cytokine expression and CRS level by CONNECTOR. a** Stability matrix for parameter G (the number of clusters) = 2, 3, 4 separately. **b** Violin plots of the fDB (functional Davies-Bouldin) calculated on each run and for different number of clusters G (left panel). Violin plots of the total tightness (T) calculated on each run and for different number of clusters G (right panel). G = 2 corresponds to the optimal scale for the fDB indexes minimization while taking the sample size into account. Dynamic changes between the mean log$_2$FC changes of the cytokines in the separate clusters (**c**) and the total cytokines (**d**) at different observation time. **e**, The graph shows the dynamic changes of CRS grade in two groups after Cilta-cel infusion. Data is represented as mean ± SEM. The *p* values are determined using Mann-Whitney U test, and those with statistical differences (*p* < 0.05) are labeled above the X axis. Source data are provided as a Source Data file.

clearly distinguish the temporal changes in CRS grading during the treatment course. The patients in Group1 experienced severe CRS which peaked around day 12, whereas the patients in Group2 having a relatively mild CRS with a low peak around day 9 (Fig. 3e). The temporal dynamics of cytokine secretion was highly consistent with CRS grade regardless of individual discrepancy.

The above results demonstrate that the timing of cytokine production aligned with the clinical symptoms, revealing a highly regular systemic inflammatory reaction upon CAR-T-tumor contact.

**Dysregulated cytokines and their clinical relevance**

In the Cilta-cel treatment, the top major adverse effects were hepatic dysfunction and coagulation dysfunction, both accounting for 92.3% with the majority being asymptomatic. Correlation analysis of inflammatory factors and end organ damage identified a constellation of cytokines was positively related to aspartate aminotransferase (AST) and alanine aminotransferase (ALT) elevation. Granzyme B and IFN-γ, known to cause liver impairment[15], showed the strongest correlation (Pearson's *r* > 0.8) (Supplementary Fig. 2a). Overall, CRS induced hepatic damage was transient and reversible since supportive care was effective in most patients.

Meanwhile, a small set of molecules, such as Angiopoietin-1 (Ang-1), PDGF-AA, PDGF-AB/BB and EGF, were noticeable to be down-regulated as CRS progressed (Fig. 2a) and inversely associated with the severity of coagulation dysfunction (Supplementary Fig. 2b). These cytokines play pivotal roles in stabilizing endothelial integrity[16,17] which was indispensable for preventing patients from coagulation system disruption[18,19]. By contrast, Angiopoietin-2 (Ang-2), an angiogenic factor, showed positive correlation with CRS grading (Fig. 2a). The ratio of Ang-2 and Ang-1, an indicator of endothelial activation[13,20,21], was responsively elevated as coagulation dysfunction turned worse (Supplementary Fig. 2b). It implied that cytokines could pose destructive effect to the endothelial steady-state condition, representing a pathophysiological feature of CAR-T induced cytokine storm.

**Immune cells-mediated signaling regulation of cytokine release**

To explore the upstream signaling responsible for cytokine regulation, transcriptome sequencing was performed in the PBMCs at different periods. Three algorithms collectively identified 10,917 differentially expressed genes (DEGs) (Supplementary Fig. 3a), which were assigned into eight clusters with consistent expression trends using soft clustering method (Fig. 4a). We noticed that clusters 2 and 7, which

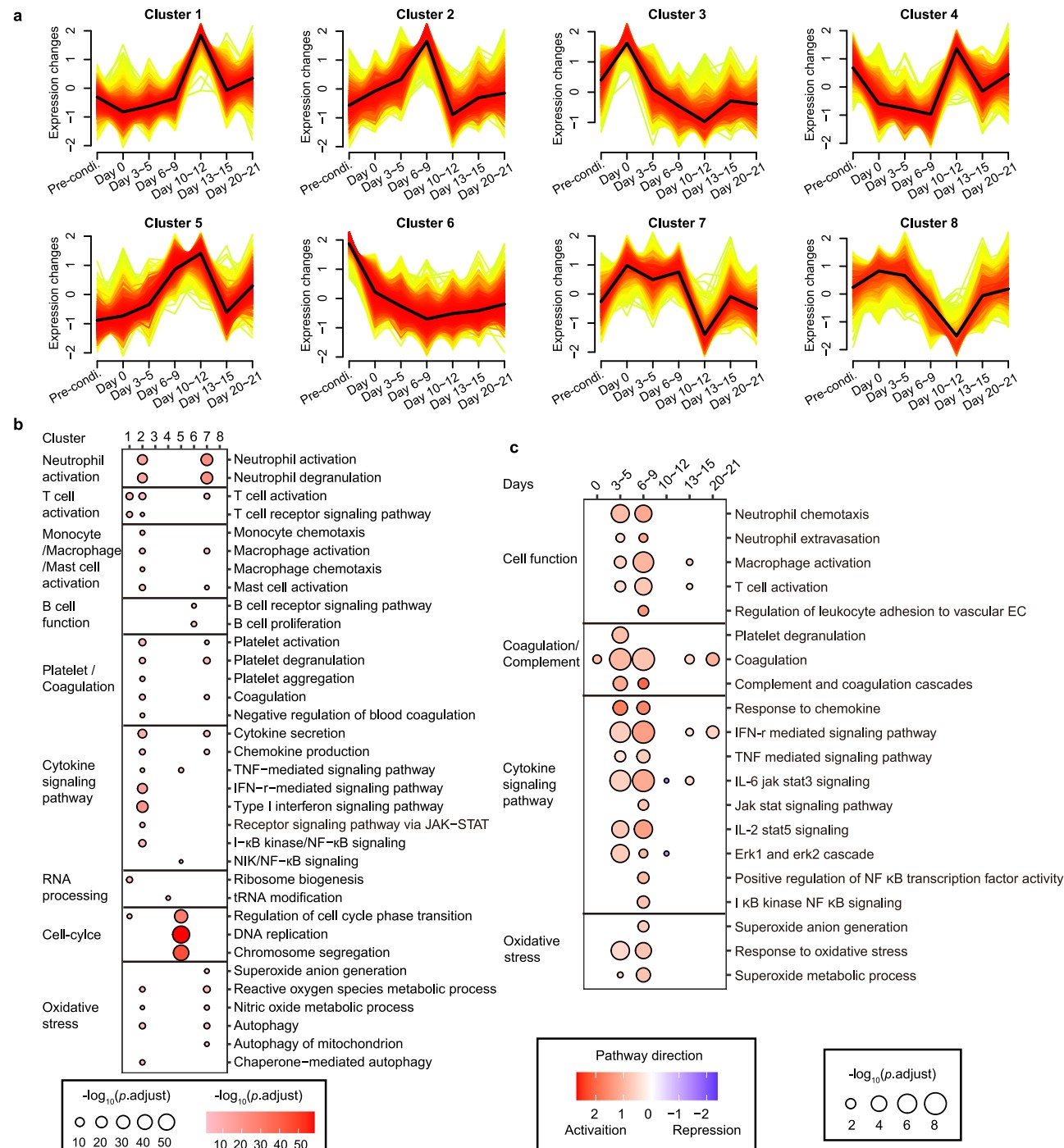

**Fig. 4 | Signaling involving CRS by RNA sequencing. a** Soft clustering of DEGs generated by transcriptome sequencing. **b** Gene ontology (GO) analyses of the biological processes indicated by core genes in the eight clusters. The dot size and darkness represent the significance of pathways in the relevant cluster (GO-computed *p*.adjust values). Pre-condi: pre-conditioning. **c** Gene set enrichment analysis

(GSEA) of RNA sequencing at indicated time slots. The dot size represents the magnitude of signaling activation (GSEA-computed *p*.adjust values). Pathway direction is the median log₂FC of significant transcripts relative to the baseline in each pathway (blue, repression; red, activation). EC indicates endothelial cell. Source data are provided as a Source Data file.

represented the signaling activated at the earlier phase (day 3–9), were significantly enriched for a variety of biological processes involved in neutrophil, T lymphocyte, platelet, monocyte/macrophage activation, cytokine signaling up-regulation, as well as excessive oxidative stress (Fig. 4a-b, Supplementary Fig. 3b). In addition, B cells affiliated genes were significantly enriched in the cluster 6, accompanied by the gene transcription being decreasingly down-regulated since lymphodepletion and Cilta-cel therapies (Fig. 4a-b). Moreover, correlation analysis showed that clusters 1 and 5, characterized by T cell activation and cell

cycle, were positively correlated with CRS severity, and cluster 7 was negatively correlated with CRS severity (Supplementary Fig. 3c). Considering CRS is a dynamic process, these data suggested that neutrophils and platelets were activated in the early stage of CRS, and so was oxidative stress. Distinctly, at the CRS aggravation stage, T-cell activation and proliferation were augmented.

To better understand the timing of immune cell activation cascade, gene set enrichment analysis (GSEA) was performed to compare each time point to the baseline. Interestingly, the neutrophils

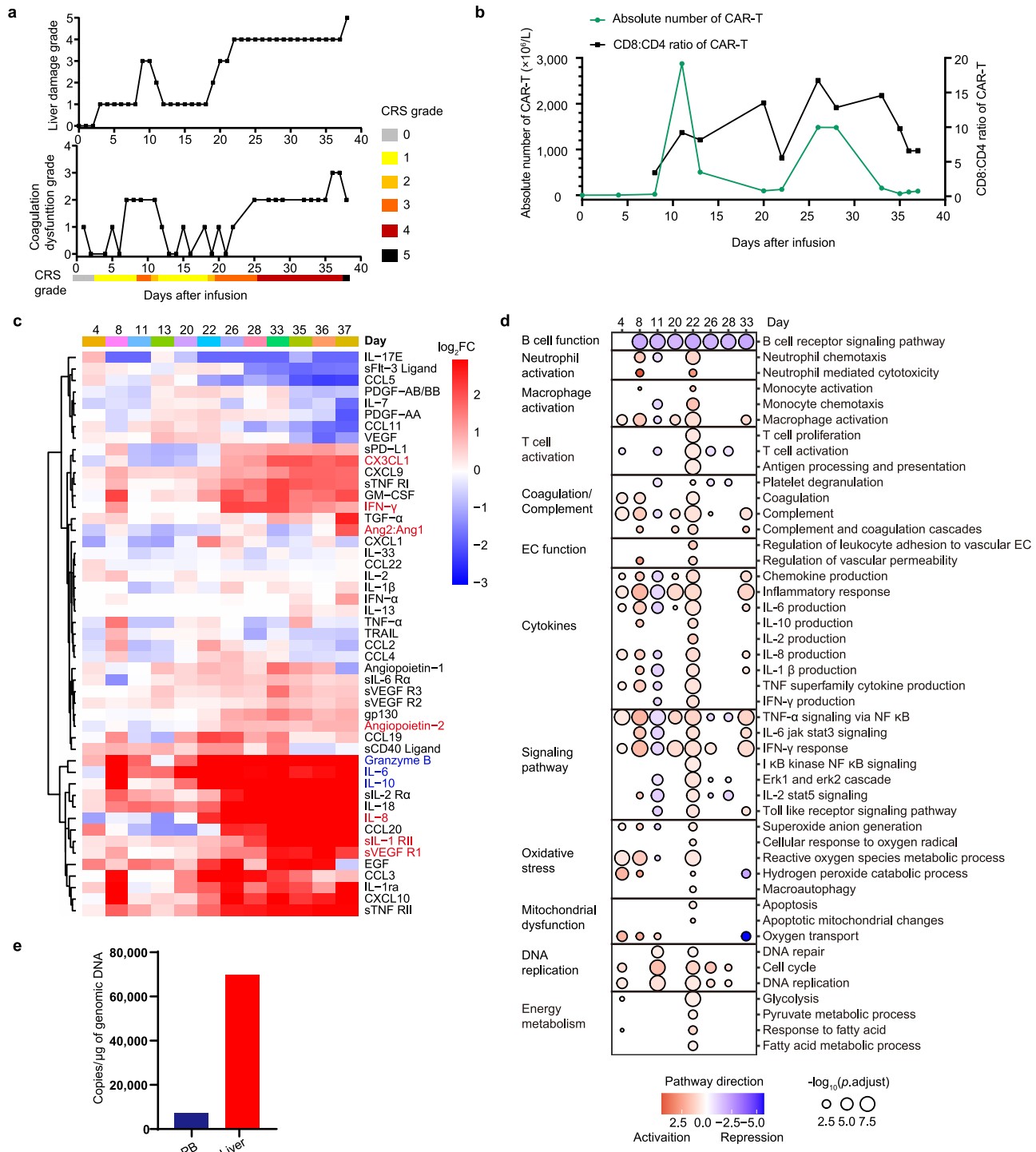

**Fig. 5 | Fatal cytokine storm in RJ-31 following the first CRS. a** The grading of adverse events (AEs) in terms of hepatic impairment and coagulation dysfunction throughout the course of double CRS in RJ-31. **b** Kinetics of circulating Cilta-cel in RJ-31. Absolute number of CAR-T is shown in green, and the ratio of CD8[+] to CD4[+] CAR-T is shown in black. **c** Cytokine profile as demonstrated by log$_2$ FC of cytokine level relative to the baseline. Ang2:Ang1 represents the ratio of angiopoietin-2 to angiopoietin-1. Cytokines with a FC value in the top three on day 8 are shown in blue. And cytokines that are significantly elevated only in the second CRS are highlighted in red. **d** GSEA of the RNA sequencing at indicated time slots. To clinically interfere with CRS, high-dose corticosteroids (20 mg dexamethasone alone or in combination with 500 mg methylprednisolone) were intravenously administered starting from day 26 as advanced supporting care couldn't effectively control the hepatic failure worsening mainly presented by the hyperbilirubinemia that rapidly progressed to grade 4. The dot size represents the magnitude of signaling activation or repression (GSEA-computed *p*.adjust values). Pathway direction is the median log$_2$FC of significant transcripts relative to the baseline in each pathway (blue, repression; red, activation). EC indicates endothelial cell. **e** The DNA copy numbers of the exogenous transgene in the liver and peripheral blood (PB) are detected by qPCR on day 38. Source data are provided as a Source Data file.

responded strongest as early as day 3-5 (latent period), while immune response from macrophages and T cells started at the latent period, and turned strong at the fever period of day 6-9 (Fig. 4c), suggesting that neutrophils acted as a prominent pioneer in initiating inflammatory reaction, whereas macrophages and T lymphocytes were more engaged in the centre of subsequent cytokine cascade. As the immune cells responded, the key inflammatory signaling pathways TNF, IL-6/JAK-STAT3, JAK/STAT, IL-2/STAT5 and complement/coagulation cascades were significantly upregulated accordingly (Fig. 4c), which were well-known for their central roles in cytokine storm[19,22–25]. Similar conclusions were drawn by gene set scores (Supplementary Fig. 4). These data were predictive of potential targets and appropriate time for the use of signaling blockers.

## CAR-T re-expansion in three patients

The majority of patients experienced CAR-T expansion once, but three cases (RJ-25, 27 and 31) presented re-expansion of Cilta-cel following the first CAR-T peak (Supplementary Fig. 5a, b, Fig. 5). Re-expansion onset time in RJ-25 and RJ-27 was half a year after CAR-T infusion whereas that in RJ-31 was only 22 days. In parallel, RJ-25 and RJ-27 had very limited cytokines produced (Supplementary Fig. 5c, d). By contrast, RJ-31 suffered life threatening CRS. Such discrepancies prompted our interest to figure out the potential causes. We found, at the half of year, RJ-25 was diagnosed with herpes zoster eye's disease, and RJ-27 was tested positive for human rhinovirus with on-going complete responses (Supplementary Fig. 5e). Viral infection in both cases was symptomatic and most likely owing to persistent hypoimmunoglobulinemia as plasma cells remained deficient (Supplementary Fig. 5f-h). Since T lymphocytes are the key guardians of adaptive immunity, it was speculated that CAR-T might took part in immune response to pathogen invasion and increasingly proliferated again. With respect to RJ-31, this patient consequently died of hepatic failure due to fatal CRS during the second CAR-T amplification. To deeply comprehend the origin of the excessive CRS in RJ-31, this special case was investigated in details.

## Case study of a fatal cytokine storm following the primary CRS

RJ-31, a 61-year-old male, presented the first severe CRS (1st_CRS) on day 9, and relieved soon after Tocilizumab administration. The adverse events (AEs) of hepatic impairment (Grade 3) and coagulation dysfunction (Grade 2) were transient during the primary immune reaction and relieved on day 12 (Fig. 5a). A second CRS (2nd_CRS) appeared on day 20 with progressing jaundice (Fig. 5a), which lasted 18 days until he died of liver failure and hepatic encephalopathy in spite of the absence of fever, hypotension at the initial stage. The Cilta-cel in the blood stream presented two peaks on day 11 and day 26-28, respectively, and the ratio of CD8+ to CD4+ circulating CAR-T was significantly higher at the second episode compared with that at the 1st_CRS (Fig. 5b, Supplementary Fig. 6a). Cytokine profile confirmed that severe inflammatory storm consecutively occurred twice in RJ-31. The 1st_CRS was manifested by a transient increase mainly in Granzyme B, IL-6, IL-10, while 2nd_CRS exhibited a durable and robust rise of more cytokines, such as IL-8, IFN-γ, CX3CL1, sIL-1 RII, sVEGF R1, Ang-2 (Fig. 5c). Compared with the common signaling activation on day 8, GSEA pathway enrichment showed a more fierce T cell immune activity, platelet degranulation and a stronger potency of various inflammatory signaling involving Erk, IKK/NF-κB, Toll-like receptor pathways and oxidative stress, mitochondrial damage on day 22, concomitant with highly active glycolysis, pyruvate metabolic and fatty acid metabolic processes (Fig. 5d). A widespread down-regulation of cytokine transcriptional signaling since day 26 might result from corticosteroids intervention (Fig. 5d).

Hepatic biopsy was obtained on day 38. The transgene copy number of Cilta-cel in the liver was 9.5 times higher than that in the matched PB of day 38 (Fig. 5e). Immunohistological staining showed a massive cytotoxic CD8+ T cell infiltration containing toxic granzyme B and TIA-1, high KP-1 (CD68) expression (Supplementary Fig. 6b-g), reflective of a T cell mediated immune response in the liver, which should be stronger than that in the PB owing to a higher accumulation of CAR-T. Since CAR-T were normally activated by the targeted antigen, we hypothesized the presence of BCMA-expressing cells may act as a stimulus. Albeit no imaging evidence to support tumor infiltration in the liver, immunohistochemistry indeed revealed a scattered distribution of plasma cells expressing CD138 and BCMA (Supplementary Fig. 6h-n), indicating that the 2nd_CRS was probably initiated in the liver with cytotoxic Cilta-cel accumulation, which was most likely relevant to the exposure of localized BCMA antigen.

## A dominant clone of Cilta-cel was identified in RJ-31 at the 2nd_CRS

Unexpectedly, RNA sequencing of PBMCs in RJ-31 uncovered a dramatic skewing of TCR repertoire to a dominance possessing TRAV13-1-J20/TRBV12-5-J2-1, which took up 89% of the entire TCR repertoire on day 28 (Fig. 6a). TCR diversity (inverse Simpson index) and clonality (clonality index) indicated that RJ-31 uniquely showed a deep reduction of TCR diversity and a remarkable increase of T cell clonality since day 22 (Fig. 6b). The other cases all showed a transient, light change of TCR diversity and clonality, followed by a quick return to the original status (Fig. 6b), reminiscent of the common findings previously described[26,27]. Lentiviral vector integration site analysis further revealed a high abundance of a clone with monoallelic integration at the downstream (nearly 14 kbp away) of ETS1 gene without affecting its transcription confirmed by quantitative PCR (Fig. 6c, Supplementary Fig. 7), ruling out the association of Cilta-cel dominance establishment and lentiviral insertion.

To functionally evaluate the property of the dominant clone, purified CAR-positive cells at the 2nd_CRS (day 28), as well as the matched control in the 1st_CRS (day 11) were harvested for single-cell RNA sequencing. Overall, CAR-T in the 2nd_CRS was enriched for the cytotoxic CD8+ effectors (Fig. 6d, Supplementary Fig. 8a-b). As a comparable control, a small dominant CAR-T clone, TRAV19-J34/TRBV4-1-J2-7 (referred to as Major_A) in the 1st_CRS was selected for comparison with the dominant TRAV13-1-J20/TRBV12-5-J2-1 (referred to as Major_B) identified 73% in the 2nd_CRS (Fig. 7e). This investigation showed that the Major_B possessed a stronger cytotoxic capacity respective of lytic cell composition (Supplementary Fig. 8c) and higher toxicity score (Fig. 6f-g, Supplementary Fig. 8d). Additionally, the significant DEGs in Major_B were involved in Toll-like receptor signaling pathway, apoptotic mitochondrial changes and calcium iron related processes (Fig. 6h), which was in line with the finding generated by PBMCs transcriptome sequencing. Meanwhile, the entire CAR-T population could be representative of the clone when the dominance took up 73%-78% during day 28-38 (Figs. 5b, 6c, e).

## HHV7 activation and TET2 mutation might be collectively responsible for clonal amplification of Cilta-cel in RJ-31

Given CAR-T clonotype rarely skewing to a single dominance upon tumor stimulation, we speculated that the dominant clone was contributed by another factor other than BCMA-expressing cells. To mine the unknown peptide, the TCR clone was shown highly matched with herpesvirus5 (HHV5) in the ImmuQuad Biotech database. Nevertheless, transcriptome sequencing of RJ-31 PBMCs surprisingly found herpesvirus7 (HHV7) but not HHV5 was activated starting from the resolving period of the first CRS. Back to the aforementioned database, HHV7 was not included as a potential target for TCR alignment indeed. Though HHV5 and HHV7 only shared 38.6% homology at peptide level, HHV7 was hypothesized as a stimulus of the CAR-T dominance formation. This hypothesis was verified by a transient positivity of the HHV7 fragment between day 20 and 26 by RT-qPCR, which was exactly

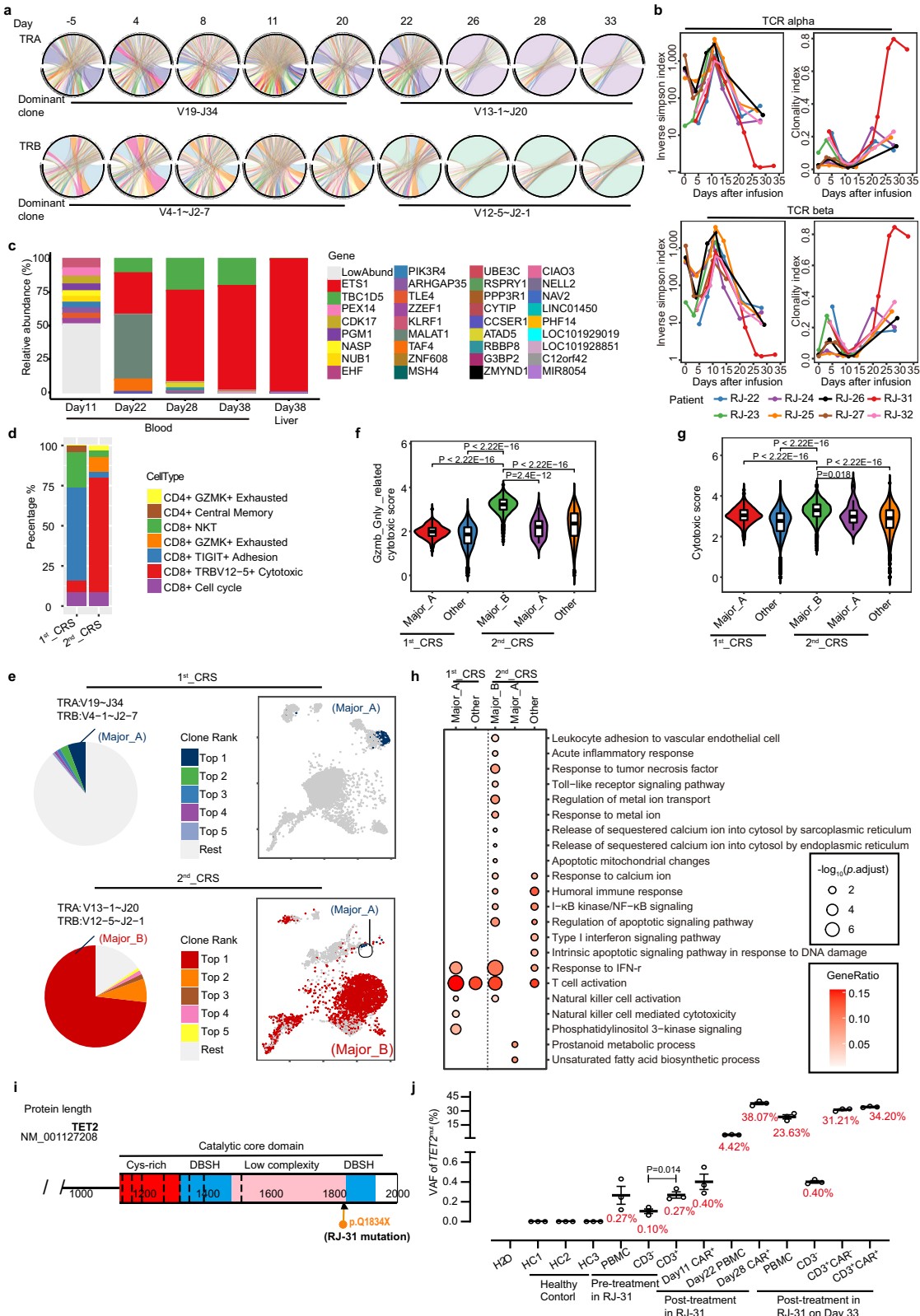

consistent with the course of 2nd_CRS (Supplementary Fig. 9a-b). Hence, *HHV7* infection was probably the key stimulus for the corresponding CAR-T clone re-expansion.

Along with the engineered cells predominantly with the CD8+ subset outgrew (Fig. 5b), CD8+ endogenous (CAR-negative) T cells preferably proliferated as RJ-31 suffered from viral infection (Supplementary Fig. 9c). It was reminiscent of the viral infection of RJ-25 and

RJ-27, both of whom also had a high CD8+ CAR-T re-expansion. At the same time point, the endogenous T cells exhibited a higher CD8+ fraction compared to that at the early treatment phase (Supplementary Fig. 9c). Taken together, the three cases shared a common feature, which was, a high CD8 CAR-T and endogenous T cell subpopulation level at virus infection. This phenomenon indicated a competent immunity against the exogenous microbe. However, we noticed that

**Fig. 6 | Identification of a dominant CAR-T clone in RJ-31. a** Circle graphs show the type of genetically VDJ rearrangement of TCR alpha chain (TRA) and beta chain (TRB). The type of dominant clone detected at certain period is shown below the graphs. **b** TCR diversity (inverse Simpson index) and T cell clonal expansion (clonality index) in eight patients after Cilta-cel treatment. **c** The abundance of integration site marks CAR-T clone in the blood and liver of RJ-31. Top ten major integration sites are shown in different colors, and the other sites with relatively low abundances are in grey. **d** Proportions of CD4 or CD8 subtypes in the purified CAR-T population at the first and second CRS strikes, respectively. **e** The pie chart shows the frequencies of the top 5 ranked clones in the purified CAR-T population. The clone with the biggest proportion is defined as a dominant clone. The clone type is characterized with a pair of specific TRA and TRB. The UMAP plots show the dominant clone distribution at the 1st_CRS (blue) or 2nd_CRS (red). **f-g** The violin charts show the cytotoxic immunophenotypes of two dominant clones respectively derived from CAR-T in the two CRSs. The annotations for the cytotoxic T subset are from our database (**f**) and those reported previously[62] (**g**) (n = 317, 5250, 4832, 19, 1756 cells following the order of the groups on the X axis). Box plots show the first quartile (lower end of the box), the third quartile (upper other end of the box) and the median value (centre line). The p values are determined using two-sided Mann–Whitney U test. **h** GO enrichment of the top DEGs in the two dominant clones (GO-computed p.adjust values). **i**, Diagram of the TET2 catalytic domain. The heterozygous point mutation (arrow) identified in RJ-31 is at the C-terminal of TET2 protein, where the catalytic domain locates. **j**, Statistical plot of the variant allele fractions (VAFs) of $TET2^{mut}$ in RJ31 (n = 3 independent experiments). HCs represent three healthy controls. Data is presented as mean ± SEM. The p value is determined using two-sided t-test. Source data are provided as a Source Data file.

the T clonotype of RJ-25 and RJ-27 kept the polyclonal state (Supplementary Fig. 5i). This triggered us to explore additional cause potentially contributing to the accumulation of a single CAR-T clone.

Interestingly, PBMCs transcriptome sequencing showed a heterozygous mutation in *TET2* gene from day 20 on. This newly identified abnormality was validated by targeted DNA sequencing, which demonstrated the variation (C > T) of *TET2* localized at the genomic site of Chr4:106197167, resulting in an early translational termination of the catalytic structural domain of TET2 protein (referred to as $TET2^{mut}$, Fig. 6i, Supplementary Fig. 10a-b). *TET2* mutation is commonly considered as an initiating event of clonal hematopoiesis[28,29], and dysfunction of TET2 protein causes an exacerbation of immune inflammation, particularly characterized by an IL-6 elevation[30]. To precisely determine if a low abundance of $TET2^{mut}$ existed before the 2nd_CRS, ddPCR was performed and revealed a small $TET2^{mut}$ clone already existed in PBMCs at pre-treatment indeed and mainly concentrated in CD3 positive host T cells (Fig. 6j, Supplementary Fig. 10c). This variation was also detected in the CAR-T on day 11 with low variant allele fraction (VAF) of 0.40%, and on day 28 with higher VAF of 38.07% by ddPCR (Fig. 6j). Moreover, in the liver biopsy on day 38 when the case RJ-31 died of the lethal toxicity, $TET2^{mut}$ was found to have similar VAF levels to that of day 28 peripheral CAR-T (Supplementary Fig. 10d). It was worth noting that $TET2^{mut}$ VAF of the CD3-negative subset was merely 0.4% at a late time point (day 33), which was basically close to the VAF (0.1%) at pre-treatment (Fig. 6j, Supplementary Fig. 10c). It suggested that the somatic $TET2^{mut}$ present at leukapheresis was more carried by T lymphocytes, consequently potentiating the engineered T cells clonal expansion, and *TET2* mutation was unable to drive non-T hematopoietic cells clonal expansion under the circumstance of microbial infection.

In sum, virus activation, a rare somatic mutation of *TET2*, along with plasma antigen stimulation in the RJ-31 liver might collectively contributed to the clonal amplification of Cilta-cel, prompting a second severe CRS and conferring an excessive inflammation, and fatal damage to the liver function.

## Discussion

This study uncovered an orchestrated chronological process of inflammation in response to CAR-T. The CRS pathophysiological processes investigated in this work are summarized in Fig. 7. We pointed out that CAR-T could be activated by either targeted tumor antigen or TCR-specific pathogen. In the common setting, CAR bound tumor cells to produce pro-inflammatory factors that initially lead to neutrophils activation, followed by participation of other immune cellular components. In a viral invasion setting, the viral peptide presented by antigen presenting cells could allow CAR-T to be activated in a classical TCR/MHC manner. Other factors, such as target antigen exposure in the malignant or normal cells might reinforce CAR-T expansion, and genetic abnormality (such as somatic mutation of *TET2*) in CAR-T might confer a proliferating advantage as well.

CRS is a result of an activation of a multicellular network. Other than T lymphocyte and monocyte/macrophage, the primary cytokine producers recognized[5–7], neutrophils were found the earliest activated after Cilta-cel infusion. As innate immune effectors, neutrophils work at the first line of response for the acute inflammation[31] and take part in the pathophysiology of CRS in COVID-19[32,33] and acute GVHD;[34] whereas, the role of neutrophils in CAR-T related CRS remains unclear. Our finding provides compelling evidence that neutrophilic immunostimulatory reaction also drives a major cellular response at the early stage of CRS outbreak in Cilta-cel therapy.

The stepwise procedure of Cilta-cel-induced CRS depicted a kinetic change in the axis of signaling-cytokine-presentation. On one hand, inflammatory signaling pathways are shown to be activated in 3-5 days prior to cytokine storm, illuminating that close monitoring and essential supporting care are needed as early as day 3-5 to reduce the risk of unmanageable toxicity caused by delayed intervention. On the other hand, the hyper-activation of IL-2/STAT5, IL-6/JAK-STAT3, TNFα signaling indicates the potential targets for drug administration. Tocilizumab is the most common agent to help control CRS[25]. TNFR blocker, Etanercept, has been applied in some cases[3]. Other agents approved for controlling GVHD, such as JAK2 inhibitor (Ruxolitinib), IL-2 receptor antagonist (Basiliximab), can also be considered used against the over-reacted inflammation in the CAR-T therapy[22,23]. Meanwhile, the optimal time point of targeted agent usage is expected not to be later than day 10, at which, both cytokines and activated signaling are reaching peak levels.

Although re-expansion of CAR-T could occur under certain circumstances[35–37], multi-factor-induced CAR-T re-expansion hasn't been elucidated yet. In this study, we report a rare case who suffered two sequential CRS hits within one month post CAR-T infusion. To our knowledge, among 219 r/r MM cases receiving Cilta-cel therapy so far[3,9,10,38], this was the only one who developed a lethal CRS with an evidenced clonal expansion of CAR-T. We assume that several factors may result in this unexpected issue: first, late immune reaction could be induced by CAR-T and antigen-exposing cells interaction again at localized lesion[39]; second, virus reactivation may occur in the setting of immune incompetency after CAR-T treatment[40,41]; third, the somatic *TET2* mutation is most likely the key intrinsic cause of CAR-T's clonal expansion and subsequent fatal inflammatory storm in RJ-31, which was not relevant to Cilta-cel manufacturing.

With respect to the somatic $TET2^{mut}$ found in this study, RJ-31 aged 61 was speculated to have the clonal hematopoiesis of indeterminate potential (CHIP), which occurs in apparently healthy elderly people in particular, with *DNMT3A*, *TET2* and *ASXL1* loss-of-function mutations being the most commonly detectable ones[42,43]. A small clone with *TET2* mutation could conceal in RJ-31's T lymphocytes before their collection for CAR-T preparation, constituting a risk of wild outgrowth of CAR-T cells derived from the autologous lymphocytes under certain circumstance, such as viral infection. Nevertheless, the other two patients (RJ-25 and RJ-27) were not detected with somatic *TET2* mutation, as well as other gene mutations associated with CHIP including

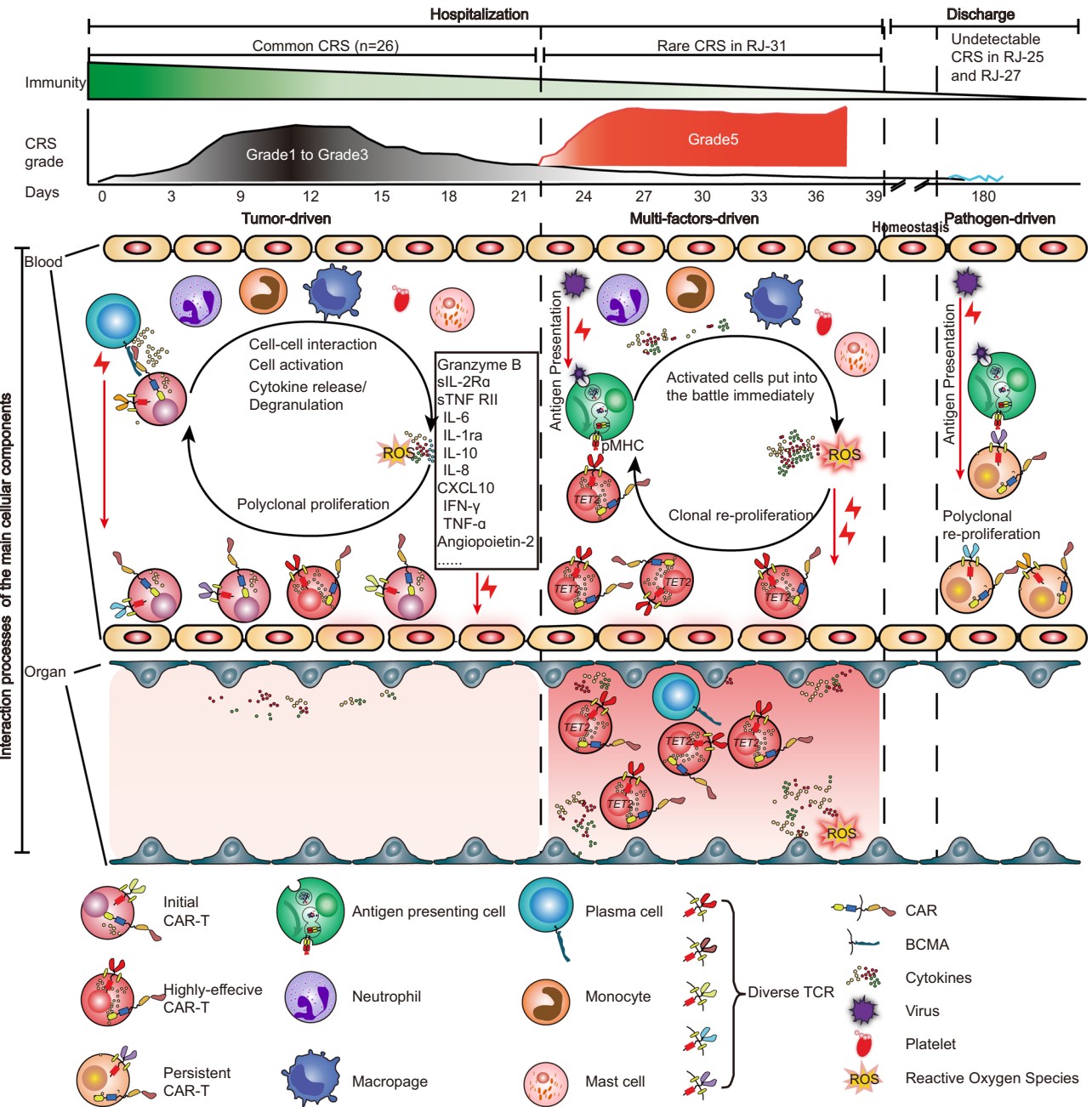

**Fig. 7 | Development of CRS after Cilta-cel therapy.** Schematic illustration shows the CRS development upon Cilta-cel therapy. Within one month after CAR-T infusion, the common CRS is driven by the contact of CAR-T with tumor antigen, followed by participation of immune cellular components (left). Under the circumstance of pathogen invasion and/or genetic mutation, activated CAR-T can be boosted up in a TCR/MHC manner, even are able to expand clonally. The immune reaction in both blood stream and tissues can be accordingly enhanced and might cause irreversible organ damage (middle). Beyond six months after infusion, low level persisting CAR-T skew to a memory-like status and are less likely to induce a robust cytokine storm against microbial infection (right).

*DNMT3A* and *ASXL1*, so the risk of TCR clonality formation was largely minimized. Therefore, for the RJ-31 case, the genetic defect was considered the overriding determinant to drive the clonal proliferation of CAR-T, and the viral infection was probably an antigen stimulus to trigger adaptive immunity with CAR-T cells being a part.

The phenomenon in RJ-31 was reminiscent of a case (Patient-10) reported in the anti-CD19 CAR-T therapy (referred to as the CD19 CAR-T case, hereafter), in which the CAR-T infused for the second time bore *TET2* sequence abnormalities on both alleles, enabling clonal proliferation of CD8[+] central memory CAR-T, and rapidly achieving tumor clearance accompanied by high-grade CRS[44]. Though the similarities in *TET2* mutagenesis, CD8[+] graft clonal outgrowth and severe toxicity, the disparity of the consequent outcomes between RJ-31 and the CD19

CAR-T case probably lies in the following reasons: first, the distinction in CAR-T immunophenotypes. The cytotoxic effector phenotype in RJ-31 was of higher cytokine-producing capacity, compared with the central memory phenotype found in the CD19 CAR-T case; second, the difference in collaborative effects. Albeit *TET2*[mut] should play a prominent role in CAR-T's aggressive growth in RJ-31, the influence from other two factors, viral activation and antigen stimulation, could not be neglected. Relatively, the CD19 CAR-T case was not reported to have other triggers besides *TET2* abnormalities; third, the difference in the CRS occurring time. The next CRS struck consecutively when RJ-31 hadn't completely recovered from the first storm, whereas the CD19 CAR-T case didn't experience sequential CRS; fourth, the distinction in cytokine profiles. The CD19 CAR-T case was observed having common

inflammatory molecules elevation; however, RJ-31's cytokine signature was considerably broadened and more toxic. Despite the objective observations above, such genetic events which happened to either RJ-31 or the CD19 CAR-T case are occasional. Screening of CHIP-associated genes, particularly *TET2, DNMT3A* and *ASXL1*, can be an optional measure in CAR-T manufacturing in some elderly populations presenting CHIP bias, such as immune cytopenia, marrow dysplasia, and nutritional deficiency.

Additionally, the life-threatening CRS in RJ-31 evoked our awareness of proper and timely administration of immunoglobulins or specific anti-microbe agent, minimizing the possibility of pathogenic infection at the period when immunoglobulin declines and circulating CAR-T remain in the effector status. Our work also conveyed an implication for the use of a bi-specific CAR-T simultaneously targeting BCMA antigen and a commonly affected virus, such as CMV. Such a combination targeted approach may help empower immunity and lower the risk of severe virus infection. Furthermore, CAR-T constitutively carrying suicide genes are in great demand to cope with life-threatening side effects.

In conclusion, our study intensively evaluated the physiological process and the timing of CAR-T therapy-related CRS development, deepening our understanding of systemic toxicities in cellular immunotherapy. Although this study was carried out in the r/r MM patients with Cilta-cel treatment, we believe that the key findings is not restricted to this typical disease type and engineered product, instead, the conclusions can be shared by some of other CAR-T treatments. Of course, CAR-T-related CRS studies with a larger sample size or distinct tumors are highly encouraged to consolidate our discoveries in the future.

## Methods

### Patients and clinical assessment

The Legend-2 (phase I) and CARTIFAN-1 (phase II) studies were two pivotal trials of Cilta-cel in China. This study included all participants with available specimens in our clinical center, Ruijin Hospital affiliated with Shanghai Jiao Tong University School of Medicine (referred to as RJ hereafter). The participating sites of the phase I trial comprised four centers with one in the West China and the other three in the East (NCT03090659, ChiCTR-ONH-17012285). The East area (RJ, First Affiliated Hospital of Nanjing Medical University and Changzheng Hospital) with RJ as the leading site administrated Cilta-cel treatment in 17 patients, of whom, 16 patients' sequential serum samples were obtained while in one case the sample was unavailable. The phase II trial was conducted in eight centers (NCT03758417). Each was in charge of its own patient samples collection. As of manuscript preparation, RJ had treated 10 phase II patients whose serum samples were entirely investigated. Therefore, 26 r/r MM patients were included in this study without any selection bias. All the evaluated patients underwent cyclophosphamide-based (with or without fludarabine) lymphodepletion therapy followed by Cilta-cel infusion at a median dose of $0.595 \times 10^6$ CAR-T cells/kg. The written informed consent to sample collection and data mining was obtained from all the patients. The study was approved by the RJ ethics committee.

Patient clinical response was assessed by the criteria of treatment response according to International Myeloma Working Group (IMWG) consensus recommendations[45]. AEs were graded according to National Cancer Institute Common Terminology Criteria for Adverse Events (NCI-CTCAE v. 4.03). Determination of CRS was based on the criteria proposed by Neelapu et al.[46]. Five grades were set for CRS stratification, in which, Grade 1–2 was defined as mild toxicity and Grade 3 to 5 was referred to as severe event. The term of coagulation dysfunction used in this study was a collective description of the AEs relevant to platelet count, activated partial thromboplastin time or fibrogen. The term of liver damage used in this study was a collective description of the AEs relevant to AST, ALT, fibrogen and Bilirubin. The grading of

coagulation dysfunction and liver damage depended on any of the above-mentioned AEs that met the end of the highest AE stage.

### CAR-T functionality assessment

The killing capacity of CAR-T was evaluated at the quality control step of manufacturing. CAR-T were co-incubated for 24 h with luciferase labeled-RPMI-8226 cells at an E:T ratio of 20:1 (phase I) or 5:1 (phase II). Controls were the untransfected T cells from the same donor. After 24 h incubation in 37°C, 5% $CO_2$ cell culture incubator, the cells in each well were added with 25 μl ONE-Glo Firefly luciferase assay reagent mix (Promega, E6120) and incubated at room temperature for 1 min. Relative light unit (RLU) of luciferase activities from the remaining living target cells were read in a microplate reader (Tecan Spark 10 M) and were used for cell viability calculation.

In phase I with an E:T ratio of 20:1, the mean apoptotic cell fraction of the mild CRS group was 98% (range: 87–100%), and that of the severe group was 95% (range: 83–100%), without significant difference. For the 10 investigated patients in phase II with an E:T ratio of 5:1, the average killing capacity of CAR-T was 61% and 56% in mild (one patient) and severe (9 patients) groups, respectively, and wasn't reflective of an evident apoptotic disparity.

### Flow cytometry

Fresh PBMCs were harvested, and then incubated with antibodies for 30 min at 4 °C for staining. CAR-T immunophenotyping was detected using fluorochrome-conjugated antibodies against human CD45 (BD Biosciences, Cat: 555483; Clone: HI30; Lot: 9337233), CD3 (BD Biosciences, Cat:555335; Clone: UCHT1; Lot: 7200698), CD4 (BD Biosciences, Cat:557871; Clone: RPA-T4; Lot: 7348612), CD8 (BD Biosciences, Cat:564116; Clone: SK1; Lot: 8137958), CD45RA (BD Biosciences, Cat:563733; Clone: HI100; Lot: 8073653), CCR7 (BD Biosciences, Cat:562555; Clone: 150503; Lot: 7355865) and FITC-labeled human BCMA protein (ACRO Biosystems, Cat: BCA-HF254; Lot: FL894-20BHF1-VK). All the data were collected using an LSRII flow cytometer (Becton Dickinson, Franklin Lakes, NJ, USA) and were analyzed with FlowJo software V10 (TreeStar, Ashland, OR, USA). Gating strategy can be found in Supplementary Fig. 11.

### Serum cytokine profiling

Magnetic Luminex Assay kits were purchased from R&D Systems (catalog number: FCSTM18-45 and LXSAHM-16). 61 interleukins, interferons, chemokines, TNFs and soluble receptors/ligands were detected and collectively called "cytokines" throughout the context. In detail, they were CCL2, CCL3, CCL4, CCL5, CCL11, CCL19, CCL20, CX3CL1, CXCL1, CXCL2, CXCL10, IL-1α, IL-1β, IL-1ra, IL-2, IL-3, IL-4, IL-5, IL-6, IL-7, IL-8, IL-10, IL-12 p70, IL-13, IL-15, IL-17A, IL-17E, IL-33, EGF, FGF basic, VEGF, PDGF-AA, PDGF-AB/BB, TGF-α, G-CSF, GM-CSF, sPD-L1, sCD40 Ligand, sFLT-3Ligand, IFN-α, IFN-β, IFN-γ, TNF-α, TRAIL, Granzyme B, sIL-2Rα, CXCL9, gp130, sIL-1R II, sIL-6Rα, sTNF RI, sTNF RII, RAGE, angiopoietin-1, angiopoietin-2, IL-18, CCL22, sBCMA, sVEGF R1, sVEGF R2, sVEGF R3. Samples cryopreserved at −80 °C were thawed and processed according to the manufacturers' protocols. Assay plates containing patient serum were placed in the Luminex X-200 instrument which provided a platform for the tests. Data processing was carried out by Milliplex analyst software based on five-parameter logistic curve-fit. All data were subject to strict quality control adhering to the following criteria: the standard curve for each analysis had a $R^2$ value > 0.99, and the coefficient of variation of standard curve and sample replicates were below 20%.

### CONNECTOR analysis

Since the CC4 and CC5 exhibit the most robust correlations with CRS severity, we adopted the computational methodology CONNECTOR to stratify patients based on their affiliation with CC4 and CC5. Utilizing the mean $\log_2FC$ as the observation, we have performed 1,000

iterations using the ClusterAnalysis function within CONNECTOR. This analysis effectively partitions the 26 patients into two distinct groups: Group1 (comprising 8 patients: CZ-02, CZ-03, JS-06, RJ-01, RJ-02, RJ-04, RJ-22, RJ-30) and Group2 (comprising 18 patients: CZ-01, JS-02, JS-03, JS-04, JS-05, JS-07, JS-08, JS-09, RJ-03, RJ-05, RJ-23, RJ-24, RJ-25, RJ-26, RJ-27, RJ-29, RJ-31, RJ-32).

## Nucleic acid extraction and RNA sequencing (RNA-seq)

Total RNA and genomic DNA were isolated from PBMCs, whole blood or liver tissue using QIAamp[R] DNA Mini Kit (Qiagen) or RNeasy Micro Kit (Qiagen). The quality of RNA/DNA was firstly verified by Agilent Bioanalyzer 2100.

RNA-seq libraries were prepared according to protocol and then sequenced on NovaSeq 6000 platform (Illumina). Raw FASTQ files were aligned to human reference genome GRCh38 (release 37). The human reference genome and its annotation file were downloaded from GENCODE database (https://www.gencodegenes.org/). We used salmon (v1.3.1)[47] to generate the count and transcripts per kilobase of exon model per million mapped reads (TPM) matrix. The transcript counts were then merged using DESeq2[48] and transformed as fragments per kilobase million (FPKM) to evaluate the gene expression level by normalizing the length of genes using the TPM matrix. To ensure the robustness of our results and identify DEGs, three well-established and recommended methods limma (v3.44.3), edgeR (v3.30.3), DESeq2 (v1.28.1) were applied. The analysis workflow for DESeq2, limma and edgeR, which were constructed based on the linear model, followed the guidelines provided in the respective software packages. Significant DEGs were identified using adjusted $p < 0.05$. And significant DEGs between any two time points were included for further soft clustering analysis. As demonstrated in Supplementary Fig. 3a, the majority of DEGs were consistently identified by all three methods, indicating a high level of confidence in our DEGs selection process. Details of analysis workflow were described in Supplementary Fig. 3a legend and "Code availability".

## Soft clustering method

The approach of soft clustering was implemented using the fuzzy c-means algorithm by the Mfuzz R software package (v2.48.0), aiming to assign the genes into different clusters in a gradual manner according to their expression values[49]. After standardization, genes were assigned a unique cluster by membership score ($\alpha$). The gene with $\alpha$ score greater than 0.6 was considered as a core gene and was used for subsequent functional enrichment analysis. Details of analysis workflow were described in Supplementary Fig. 3b legend and "Code availability".

## Gene ontology (GO) analysis and gene set enrichment analysis (GSEA)

The functional enrichment analysis of GO was performed with clusterprofiler R package (v3.16.1). And GSEA was conducted based on molecular signatures database v7.4 and GSEABase R package (v1.50.1). Gene sets used in GSEA were downloaded from the Molecular Signatures Database (MSigDB, v7.4) of the Broad Institute.

The genes used for GSEA were selected from the FPKM profile. Genes with low expression (expressed in less than 30% of the samples) were excluded. During the GSEA analysis at various observation time points, genes were pre-ranked based on their $\log_2$FC, which was calculated by comparing their expression levels with the baseline. Pathway direction is the median $\log_2$FC of significant transcripts (core enrichment genes) relative to the baseline. In this visualization, an adjusted $p < 0.05$ with pathway direction $> 0$ indicated activation, while an adjusted $p < 0.05$ with pathway direction $< 0$ indicated repression.

## T cell receptor repertoire analysis

The inverse Simpson index (1/D)[50] and clonality index[51] were calculated as reported to quantitatively define the diversity of T cell clones.

The amino acid sequences of CDR3 localized on TRB and TRA were in a perfect match with the TCR-antigen database developed by Immuquad[@] biotech, respectively. The screening method was carried out based on the algorithm of Levenshtein distance[52] $\leq 3$, along with V and J genes exactly matched with the ones in the database.

## Virus sequences alignment and preprocessing

We have download 13,778 virus genomes from National Center for Biotechnology Information (NCBI, https://ftp.ncbi.nlm.nih.gov/) and built an in-house viral database. To reduce the impact of genomic homology between human and virus, a human-virus genome was built and we used two methods STAR (v2.7.9a)[53] and salmon (v1.3.1)[47] to calculate the virus transcripts from RNA-seq data.

## Mutations calling and screening

To identify mutations from RNA-seq data, raw reads were aligned to human reference using STAR (v2.7.9a)[53], which was downloaded from UCSC Genome Browser (http://hgdownload.soe.ucsc.edu/). SAMtools (v1.8-47) was used to generate name-sorted and indexed BAM files and the PICARD tool (https://broadinstitute.github.io/picard/) was used to mark duplicate reads in BAM files. The mutation calling steps followed the Genome Analysis Toolkit (GATK, v4.1.8)[54] forum recommended best-practice pipeline. The steps were described as followed: (i) two software, GATK HaplotypeCaller and VarScan2 (v2.4.4), were used to generate mutations including single-nucleotide variations (SNVs) and small insertions/deletions (INDELs); (ii) mutations were merged together and annotated by ANNOVAR[55] using specific databases such as snp138, avsnp147, avsnp150, COSMIC, SIFT and PolyPhen. (iii) RNAmut was used for verification mutation sites from RNA-seq data; (iv) the filtration conditions were $\geq 10x$ depth in the variant sites, $\geq 3\%$ variant allele frequency (VAF) and $> 3$ individual mutant reads, not a SNP and annotated by COSMIC database or predicted as damaging by ANNOVAR.

## Lentiviral vector integration site analysis

Integration site analysis was implemented based on standard ligation target amplification PCR and next-generation sequencing. Briefly, 500 ng of genomic DNA was sheared to an intermediate length of $500 \pm 50$ bp using the Covaris M220 Gene Fragmentation Instrument. After purification, the sheared DNA was extended by biotinylated 3'-long terminal repeat (3'-LTR)-specific primer (Primer 1) and captured by streptavidin-coated beads (Dynabeads™ M-280; invitrogen). The captured genome was seeded to a ligation cassette including a sequencing barcode (UMI) according to the kit (Fast-Link Ligation Kit; Epicentre). Amplification of integration site was conducted via a procedure of two nested PCR. The first exponential PCR primers were referred to as Primer 2 and Primer 3. After magnetic capturing, the product underwent the second exponential PCR using Primer 4 (PE 2.0 UMI short). Primer sequences are summarized in the Supplemetary Table 1.

The purified product proceeded to the subsequent Illumina Miseq next-generation sequencing platform for paired-end reads with parameter settings referenced in the literature[56,57]. Raw sequencing data were analyzed with Genome Integration Site Analysis Pipeline[58] based on human reference genome GRCh38.

## Single-cell RNA sequencing (scRNA-seq)

CAR-positive cell populations from RJ-31's PBMCs on day 11 and day 28 were obtained by flow cytometry cell sorting (FACSAria™ III BD Biosciences) with FITC-labeled human BCMA protein (ACRO Biosystems). The target cells underwent single cell isolation to form Gel Bead in Emulsion by 10x Genomics Chrominum™. And then scRNA-seq

libraries were performed according to the Chromium Single Cell 5′ Library Construction Kit and Chromium Single Cell Human TCR Amplification Kit (10x Genomics). After quality verification, the libraries were sequenced on NovaSeq 6000 platform (Illumina).

The raw sequencing reads in FASTQ format of single-cell RNA sequencing (scRNA-seq) data were aligned to the human GRCh38 reference (2020-A version) and the gene expression matrices were generated using CellRanger (10X Genomics, default settings, Version 6.0.2). Both CellRanger software and the reference were downloaded from 10X Genomics website (https://www.10xgenomics.com/). And for V(D)J T-cell analysis, the single-cell T cell receptor (TCR) sequencing (scTCR-seq) data were aligned to human GRCh38 (ensembl-5.0.0) using the vdj module in CellRanger. The gene expression matrices generated by CellRanger were then imported and merged using the Seurat R package[59] for downstream data analysis. Cells that expressed less than 800 genes or over 10% mitochondrial RNA were filtered out. Cells which matched with TCR were identified as T cells and for further study. Top 3,000 highly variable genes were identified to perform principal component analysis (PCA) and ComBat method in R package SVA[60] was used to correct batch effects. UMAP (uniform manifold approximation and projection) was chosen to further cell visualization. Marker genes of each cluster were calculated using 'FindAllMarkers' function in Seurat. Cell types were identified and annotated by the following criteria: (i) annotation generated in SingleR[61] was used as reference, (ii) manually annotated using the top expressed genes in each cluster, (iii) clone type identified by scTCR-seq.

To compare the toxicity capacity of cells, we calculated the average expression level of toxicity-related genes for each cell to quantify cytotoxicity, called cytotoxic score. Based on our own scRNA-seq data, we computed the Pearson correlation coefficients of all genes with *GZMB* and *GNLY* separately. And then intersection of the top 20 ranked genes significantly associated with *GZMB* and *GNLY* were defined as Gzmb_Gnly_related gene set. Meanwhile, we extracted the reported toxicity-related gene set[62] as a supplement.

**Quantitative real-time polymerase chain reaction (qPCR)**
To measure the copy number of the transgene in the Cilta-cel, qPCR of targeted fragment was subsequently performed as reported previously[3].

And the transcriptional levels of the gene encoding herpesvirus7 (*HHV-7*) and *ETS1* were measured by RT-qPCR. 500 ng of RNA was reverse-transcribed into cDNA using HiScript III RT SuperMix kit (Vazyme), and ChamQ SYBR qPCR Master Mix kit (Vazyme) was applied for qPCR of the genetic segment of interest. The procedure was run and analyzed on ViiA™ 7 system (Life technologies). Gene transcriptional level was assessed using cycle threshold and normalized to a housekeeping gene *GAPDH*. The sequences of the primers were as follows: *GAPDH* forward 5′-GTCTCCTCTGACTTCAACAGCG-3′ and reverse 5′-ACCACCCTGTTGCTGTAGCCA-3′, *HHV-7* encoded gene forward 5′-AGAGGTTGCTTCTTCGAGTCA-3′ and reverse 5′-GCGCTTGTCAAAATCCTCAAAT-3′, *ETS1* encoded gene forward 5′-GGAATGTGCAGATGTCCCAC-3′ and reverse 5′-CATTCACAGCCCACATCACC-3′.

**Sanger sequencing**
The PCR was performed using Phanta Max Super-Fidelity DNA Polymerase kit (Vazyme, P505-d1). The sequences of the primers of *TET2* were forward 5′-GTCTCAGCCGATGGATCTGTA-3′ and reverse 5′-ATCTGTTGTAAGGCCCTGTGA-3′.

**Targeted DNA sequencing**
Targeted DNA sequencing was performed on the gene coding sequence. Library enrichment was carried using NadPrep DNA Preparation Kit (for Illumina) and then sequenced on NovaSeq 6000 platform (Illumina). Variant calling from targeted DNA sequencing was performed as reported previously[63].

**Droplet digital PCR (ddPCR)**
To identify the low-frequency mutation in *TET2*, ddPCR was performed in RJ-31's genome DNA at different time points. To make the assay more convinced, water and three healthy controls were included as blank and negative controls, respectively. ddPCR was accomplished on a Bio-Rad QX200 system using ddPCR supermix for Probe (No dUTP) (Bio-Rad) and analyzed on QuantaSoft (Version 1.7.4). The primers and probe sequences targeting *TET2* were forward, 5′-GTCCAAG-GAGGCTTACACAAAT-3′; reverse, 5′-ACCTCATCGTTGTCCTCTGC-3′; wild probe, 5′VIC-CACACCCTGGACTAG-3′MGB; mutate probe 5′FAM-CACACCCTAGACTAGT-3′MGB. To ensure the quality of the experiment, three replicates were completed for each sample, and each reaction was guaranteed to have a droplet count of more than 10,000. $\text{VAF} = \frac{\text{Concentrate of mutate}}{\text{Concentrate of mutate and wild}}$

**Statistical analysis**
Data analyses were performed using R version 4.0.3 or software GraphPad Prism version 8.0.2. Statistical tests were described in the figure legends.

**Reporting summary**
Further information on research design is available in the Nature Portfolio Reporting Summary linked to this article.

## Data availability
Raw sequencing data generated during this study have been deposited in the Genome Sequence Archive in National Genomics Data Center, China National Center for Bioinformation/Beijing Institute of Genomics, Chinese Academy of Science (https://ngdc.cncb.ac.cn/gsa-human), with accession number 'GSA-Human: HRA005381' [https://ngdc.cncb.ac.cn/gsa-human/browse/HRA005381]. These data are under controlled use conditions set by human privacy regulations, only available upon reasonable request. Access can be obtained by approval via the Data Access Committee of the GSA-human database (https://ngdc.cncb.ac.cn/gsa-human/document/GSA-Human_Request_Guide_for_Users_us.pdf). Original data for graphs is provided in the Source Data file. Source data are provided with this paper.

## Code availability
Code used in the study is available on GituHub (https://nrctm-bioinfo.github.io/MM_CART_project/index.html) and has been assigned a (https://doi.org/10.5281/zenodo.10200875).

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

## Acknowledgements

This work was supported by the State Key Laboratory of Medical Genomics, the Double First-Class Project (WF510162602) from the Ministry of Education; the Shanghai Collaborative Innovation Program on Regenerative Medicine and Stem Cell Research (2019CXJQ01); the Overseas Expertise Introduction Project for Discipline Innovation (111 Project;B17029); the National Natural Science Foundation of China (Nos. 82230006, 81970189, 82070227); the Shanghai Clinical Research Center for Hematological disease (19MC1910700); the Shanghai Shenkang Hospital Development Center (SHDC2020CR5002, SHDC2020CR2066B); the Shanghai Major Project for Clinical Medicine (2017ZZ01002); the Innovative Research Team of High-level Local Universities in Shanghai; the Shanghai Pujiang Program (No. 22PJ1409800).

## Author contributions

S.-J.C., Z.C., J.-Q.M. and S.Y. designed the project. J.-Q.M., J.X., S.J., L.C., J.H. performed the clinical trial. S.Y., C.L., X.M., Y.S. performed the experiments. S.Y., Y.D., J.L. provided the bioinformatics analysis. S.-J.C., Z.C., J.-Q.M., S.Y., J.X. supervised the study and wrote the manuscript. All authors have read and approved the final version of this manuscript.

## Competing interests

The authors declare no competing interests.
