## [Peer Review File · Nature Communications]

Neutrophil activation and clonal CAR-T re-expansion underpinning cytokine release syndrome during ciltacabtagene autoleucel therapy in multiple myelomaREVIEWER COMMENTS

Reviewer #2 (Remarks to the Author):

In the manuscript entitled 'Investigation of cytokine release syndrome and a severe clonal CAR-T re-expansion in ciltacabtagene autoleucl therapy for multiple myeloma', Shuangshuang Yang et al. examine serum cytokines and circulating immune cell transcriptomes in 26 patients with relapsed/refractory (r/r) multiple myeloma (MM) after treatment with ciltacabtagene autoleucl (cilta-cel). This thorough longitudinal analysis revealed for the first time a stepwise evolution of the cilta-cel-mediated acute inflammation and suggested that neutrophil activation served as the initial signal of a beginning cytokine release syndrome (CRS). Additionally, careful post-infusion immune monitoring showed CAR-T cell re-expansion in a subset of patients, among them one bearing a somatic TET2-mutation who developed a fatal cytokine storm. Notably, an in-depth case study of this patient revealed a dramatic clonal expansion of cytotoxic effector CAR-T cells, broad cytokine profiles, and irreversible hepatic toxicity, all of which coincided with a concomittant viral infection (HHV7). Additionally, the results emphasize the need of a systematic anti-infectious prophylaxis and close monitoring of the circulating CAR-T-cell- and cytokine-levels in the peripheral blood to prevent lethal risks.

Major comments:

1) The specimen were obtained from 26 r/r MM patients receiving cilta-cel in the context of the phase I Legend-2 or Phase II CARTFAN-1 trial. Please provide additional information for the rationale of the patient selection and whether these were consecutive patients.

2) Some of the patients received only Cy-based lymphodepletion. Did this have an impact on the toxicity profile?

3) Figure 1a shows the experimental design and illustrates that serum samples were obtained from 26 patients and purified PBMCs only from a subset of those patients. PBMC were assigned to three experimental groups. To enhance the clarity of the work, it would be helpful to provide the 12 patient IDs of the flow cytometry group, the 9 patients in the RNASeq group.

4) In the result section (page 6 and Figure 1), the authors separate the clinical presentation of the CRS into 5 slots including baseline (before infusion), latent (day 3-5), fever (day 6-9), acute aggravation (day 10-15), and resolving (day 20-21). However, there is an additional group in Figure 2 (Day 10-12 and Day 13-15). Please provide a rationale for the discrepancy.

5) Changes in the Angiopoietin-1 versus Angiopoietin-2 levels have been described previously in the setting of CD19 CAR-T cell therapy. It would be important to refer to this work, for example by K.A. Hay et al, Blood 2019; 133(15):1652-1663.

6) The authors state that the cells bearing the TET2-mutation were enriched in the cytotoxic CD8+ CAR T-cell subset using single cell RNA Seq anaylsis on a single time point. Were PBMC of this patient also examined by flow cytometry (i.e. in Extended Data Fig 1 c-d). Are sequential flow cytometry data available of the immune phenotype of this immune-dominant clone ?

7) The results of this work may have implications for the use of bi-specific CAR-Ts (i.e. CMV-specific & BCMA-redirected) in the treatment of malignancies. Please discuss this in the discussion section.

Reviewer #3 (Remarks to the Author):

The current study comprehensively evaluated the timing of CAR T cell related CRS development, and systemic toxicities associated with treatment. In 26 patients, the cytokines and PBMC transcriptomics were analyzed, as well as TCR repertoire analysis in a subset of patients. Noteworthy, the work identifies neutrophilic activation in driving early CRS responses following BCMA-CAR T cell therapy. TET2 somatic mutation and viral infection post CAR T cell therapy may

drive re-expansion of CAR T cells in 3 patients promoting a 2nd wave that caused lethal CRS, with TET2 somatic mutation and viral clonality suspected to be the driver in the former case. This work is clearly detailed and performed at a high technical level. Some of these findings have been described with CD19-CAR T cells, but the impact of neutrophils and the lethal cases that were interrogated are of significance to the field. Several questions should be addressed:

1. It is unclear whether CRS level correlates with some of these readouts, including cytokine profiles and RNA sequencing performed in Fig 1 and 2.
2. Was pre-treatment or leukapheresis indicative of immune cell frequency shifts that are predictive to the subsequent CRS in patients?
3. Was the CAR T cell product functionality evaluated? Is that predictive to the subsequent CRS in patients?
4. Do the prior reports correlate therapeutic response to CRS? It is not included in this work on a per patient basis.
5. If viral infection is the culprit in the other lethal cases, why is TCR clonality not impacted?
6. Are CAR T cells and endogenous T cells reactive to virus in those cases?
7. What is TET2 VAF % in CD3-negative populations at later times.

Reviewer #4 (Remarks to the Author):

The work proposed by the authors is interesting and allow inspecting the overall landscape of the cytokine profile in those patients reporting one of the most common adverse effect of the cytokine release syndrome after the CAR-T therapy. In the following, I report some crucial points the authors should be considered.

The cytokine profile is analyzed without considering the time properly. Specifically, the heatmap of Figure 1C is difficult to interpret since the reader completely lost the temporal evolution of the data. The comments reported are not accurate. For example: on page 6 from line 10 to line 12 it is not completely true since the upregulated molecules do not follow the CRS grade for all the subjects.

I strongly suggest using some computational approaches designed to specifically analyze longitudinal data, as well as the framework published by Pernice et al doi: 10.1093/bioinformatics/btad201

Using methodology as CONNECTOR could allow you to stratify the patients according to the cytokine expression and then identify a relation among cytokines, CRS, and specific groups of patients.

Moving to the RNASeq analysis. The authors did not specify which is the design formula used in the differential expression analysis of DESeq2. Since in DESeq2 it is necessary to define two groups of comparison, how do the authors define these groups?

Moreover, the authors report that the DE genes were also calculated from the other two packages limma and edgeR, also in this case it is not specified how the design matrix is defined. This is an important point: generally, limma and edgeR are both based on an ANOVA analysis (based on the study of the variance) so why did the authors decide to use two different libraries that used the same approach, aka linear model?

For what concern the design matrix, since it is possible to design a higher order time series matrix, which design matrix was implemented by the authors?

At the end of the definition of DE genes from all three packages, how do the authors select the genes for the GSEA analysis? The intersection of the DE genes obtained from the three methods?

Does the analysis of the time series data by soft clustering method consider all genes or only the DE genes?

REVIEWER COMMENTS

Reviewer #2 (Remarks to the Author):

In the manuscript entitled ‘Investigation of cytokine release syndrome and a severe clonal CAR-T re-expansion in ciltacabtagene autoleucel therapy for multiple myeloma’, Shuangshuang Yang et al. examine serum cytokines and circulating immune cell transcriptomes in 26 patients with relapsed/refractory (r/r) multiple myeloma (MM) after treatment with ciltacabtagene autoleucel (cilta-cel). This thorough longitudinal analysis revealed for the first time a stepwise evolution of the cilta-cel-mediated acute inflammation and suggested that neutrophil activation served as the initial signal of a beginning cytokine release syndrome (CRS). Additionally, careful post-infusion immune monitoring showed CAR-T cell re-expansion in a subset of patients, among them one bearing a somatic TET2-mutation who developed a fatal cytokine storm. Notably, an in-depth case study of this patient revealed a dramatic clonal expansion of cytotoxic effector CAR-T cells, broad cytokine profiles, and irreversible hepatic toxicity, all of which coincided with a concomitant viral infection (HHV7). Additionally, the results emphasize the need of a systematic anti-infectious prophylaxis and close monitoring of the circulating CAR-T-cell- and cytokine-levels in the peripheral blood to prevent lethal risks.

Major comments:

1) The specimen were obtained from 26 r/r MM patients receiving cilta-cel in the context of the phase I Legend-2 or Phase II CARTFAN-1 trial. Please provide additional information for the rationale of the patient selection and whether these were consecutive patients.

Answer: The reviewer’s comment is right. We’re sorry not clearly elucidating the sample source. This study included all subjects with available specimens in our clinical center, Rui Jin Hospital affiliated with Shanghai Jiao Tong University School of Medicine (referred to as RJ hereafter). The participating sites of the phase I trial comprised four centers with one in the West China and the other three in the East. The East area with RJ as the leading site administrated Cilta-cel treatment in 17 patients, of whom, 16 patients’ sequential serum samples were obtained while in one case the sample was unavailable. The phase II trial was conducted in eight centers. Each was in charge of its own patient samples collection. As of manuscript preparation, RJ had treated 10 phase II patients whose serum samples were entirely investigated. Therefore, 26 patients composed of 16 from phase I and 10 from phase II were investigated in this study without any selection bias. We have added the above information to the revised manuscript (**Line 462 ~ 473**).

2) Some of the patients received only Cy-based lymphodepletion. Did this have an impact on the toxicity profile?

Answer: The reviewer’s question is important. In this study, 8 patients received Cy single-drug preconditioning and 18 received Cy+Flu combination scheme. By Fisher’s exact test, we found patients in Cy+Flu group experienced more severe CRS (grade>2) than those in Cy group (15/18 versus 2/8, p=0.008). This result is consistent

with the observation previously stated (Hay, K.A., et al, Blood, 2017). We have thus added a table assigned Supplementary Data 1 to detail the above information (**Line 109~114**).

3) Figure 1a shows the experimental design and illustrates that serum samples were obtained from 26 patients and purified PBMCs only from a subset of those patients. PBMC were assigned to three experimental groups. To enhance the clarity of the work, it would be helpful to provide the 12 patient IDs of the flow cytometry group, the 9 patients in the RNASeq group.

Answer: We thank the reviewer for the constructive suggestion. We set up a new table numbered Supplementary Data 1 as above mentioned, in which all the assessments carried out in this study were correspondingly listed under each patient's identity number.

4) In the result section (page 6 and Figure 1), the authors separate the clinical presentation of the CRS into 5 slots including baseline (before infusion), latent (day 3-5), fever (day 6-9), acute aggravation (day 10-15), and resolving (day 20-21). However, there is an additional group in Figure 2 (Day 10-12 and Day 13-15). Please provide a rationale for the discrepancy.

Answer: We appreciate the reviewer's pertinent concern and feel sorry for not presenting the data clear enough. Based on the CRS grading, the treatment course was divided into 5 slots including the Day 10-15 which was defined as the acute aggravation (AA) period. This period featured robust cytokine secretion. In order to intensify cytokine information, we collected individual samples two times (Day 10-12 and Day 13-15) within the AA period. The results of Day 10-12 and Day 13-15 were separately shown but collectively reflected the status of cytokine storm since the cytokine expression signature (Figure 1c) of Day 10-12 was similar to that of Day 13-15. In alignment with the serum analysis, RNA sequencing was performed with the PBMCs of the same time points (Figure 2a-c). For a better understanding and visualization, we have updated **Figure 1b and 1c** for the annotation of 5 periods, with the Day 10-12 and Day 13-15 being respectively denoted as AA1 and AA2. We have explained the assignment rationale of day 10-15 time slot in the revised manuscript (**Line 131~136**).

revised Figure 1b

revised Figure 1c

5) Changes in the Angiopoietin-1 versus Angiopoietin-2 levels have been described previously in the setting of CD19 CAR-T cell therapy. It would be important to refer to this work, for example by K.A. Hay et al, Blood 2019; 133(15):1652-1663.

Answer: We thank the reviewer for reminding us of the important literature referring to Angiopoietin-1 and Angiopoietin-2 (K.A. Hay et al, Blood 2019; 133(15):1652-1663 and K.A. Hay et al, Blood 2017; 130, 2295-2306). These references have been cited in the Results section (**Line 182**).

6) The authors state that the cells bearing the TET2-mutation were enriched in the cytotoxic CD8+ CAR T-cell subset using single cell RNA Seq analysis on a single

time point. Were PBMC of this patient also examined by flow cytometry (i.e. in Extended Data Fig 1 c-d). Are sequential flow cytometry data available of the immune phenotype of this immune-dominant clone ?

Answer: Yes! Immunophenotyping of the PBMCs from the *TET2* mutation-bearing patient was carried out by flow cytometry. Since single-cell TCR sequencing combined with lentiviral vector integration site analysis indicated the percentage of this monoclonal cellular component in CAR-T ranged from 73% to 78% during day 28~38 (Fig.4c, 4e), we roughly assumed that the CAR-T population at day 28~38 could be representative of this immune-dominant clone. The flow cytometry result showed that the ratio of the CD8 to CD4 of the CAR-T rose as CRS occurred and reached a peak level at 28~33 days after infusion (Fig.3b, the representative flow cytometric plot was shown below, as well as in the **revised Supplementary Fig. 7a**), reflecting the cytotoxicity phenotype of this dominant clone. We have added the above validation assay data to the revised manuscript (**Line 246~249, 301~302**).

revised Supplementary Fig. 7a

7) The results of this work may have implications for the use of bi-specific CAR-Ts (i.e. CMV-specific & BCMA-redirected) in the treatment of malignancies. Please discuss this in the discussion section.

Answer: We agree with the reviewer's perception. Indeed, virus-specific T cell has been applied in patients, particularly those vulnerable to virus activation after allogeneic hematopoietic stem cell transplantation. Likewise, CAR-T cells against plasma cells can result in compromised humoral immunity due to hypogammaglobulinemia. Reprogramming T cells to simultaneously target BCMA antigen and a commonly affected virus, such as CMV, is definitely a novel approach to empower immunity and lower the risk of severe virus infection. Such a modality is accessible to other diseases treated with different CAR-T products as well. We have added the above to the revised Discussion section (**Line 445~448**).

Reviewer #3 (Remarks to the Author):

The current study comprehensively evaluated the timing of CAR T cell related CRS development, and systemic toxicities associated with treatment. In 26 patients, the cytokines and PBMC transcriptomics were analyzed, as well as TCR repertoire analysis in a subset of patients. Noteworthy, the work identifies neutrophilic activation in driving early CRS responses following BCMA-CAR T cell therapy. *TET2* somatic mutation and viral infection post CAR T cell therapy may drive

re-expansion of CAR T cells in 3 patients promoting a 2nd wave that caused lethal CRS, with TET2 somatic mutation and viral clonality suspected to be the driver in the former case. This work is clearly detailed and performed at a high technical level. Some of these findings have been described with CD19-CAR T cells, but the impact of neutrophils and the lethal cases that were interrogated are of significance to the field. Several questions should be addressed:

1. It is unclear whether CRS level correlates with some of these readouts, including cytokine profiles and RNA sequencing performed in Fig 1 and 2.

Answer: We appreciate the critical comment from the reviewer. In this study, we included patients with different CRS levels and conducted comprehensive longitudinal profiling. With respect to cytokine profile, as illustrated in Figure 1c, the existing data showed that CRS severity was positively correlated with the up-regulation of sIL2R α , sTNF RII, and Granzyme B, and was conversely correlated with the down-regulation of Ang-1, PDGF-AA, PDGF-AB/BB and EGF. Apart from these factors, we included all the other molecules to expand the examination of mean CRS grade correlation. Additional data are supplemented in the revised version (**revised Fig.1c**, Supplementary Data 2). To provide a holistic view of the relevance between CRS level and cytokine profiles, we employed unsupervised clustering to categorize the 61 cytokines into distinct clusters. In general, we identified five cytokine clusters (CCs), among which, CC3, CC4, and CC5 displayed a positive correlation with mean CRS grade, whereas CC2 exhibited a negative correlation ($P < 0.05$, revised Figure 1c, **revised Supplementary Fig. 2b**). Notably, among the cytokines positively correlated with CRS severity, sIL2R α and sTNF RII (both belonging to CC4) and Granzyme B (belonging to CC5) exhibited the strongest correlations.

For the signaling pathways generated by RNA-sequencing, we employed soft clustering of genes using the 'mfuzz' package (as shown in Figure 2b), and identified 8 distinct gene clusters. Correlation analysis showed that clusters 1 and 5, characterized by T cell activation and cell cycle, were positively correlated with CRS severity. The cluster 7, characterized by early activation of neutrophil, platelet and oxidative stress, was negatively correlated with CRS severity (**revised Supplementary Fig. 4c**). Considering CRS is a dynamic process, these data suggested that neutrophils and platelets were activated in the early stage of CRS, and so was oxidative stress. Distinctly, at the CRS aggravation stage, T-cell activation and proliferation were augmented.

The above data have been added to the revised manuscript (**Line 143~149 and 199~205**).

revised Figure 1c

revised Supplementary Figure 2b

revised Supplementary Figure 4c

2. Was pre-treatment or leukapheresis indicative of immune cell frequency shifts

that are predictive to the subsequent CRS in patients?

Answer: We thank the reviewer for raising the important question. To this end, we performed the analyses from two perspectives. First, we analyzed the peripheral host immune cells count at pre-treatment. The average frequencies of neutrophil and lymphocyte, and monocyte in the 9 patients with mild CRS (grade \leq 2) were respectively $2.61\times 10^9/L$, $0.42\times 10^9/L$ and $0.22\times 10^9/L$, and those in the rest 17 with severe CRS (grade $>$ 2) were respectively $2.08\times 10^9/L$, $0.15\times 10^9/L$ and $0.14\times 10^9/L$. To our surprise, endogenous lymphocyte count at baseline was significantly lower in the severe CRS group than that in the mild group ($p=0.016$). However, there was no statistical difference in the neutrophils ($p=0.545$) and monocyte ($p=0.199$) counts between the mild and severe CRS groups. Second, we collected the information of engineered and host T cells composition at pre-treatment. 16 patients had the available data of CD8 and CD4 frequencies in CAR-T products. The result showed that the mean proportions of CD8⁺ and CD4⁺ CAR-T cells in the mild CRS group were respectively 59% and 41%, and those in the severe group were respectively 62% and 38%. Neither CD8⁺ nor CD4⁺ reached significant difference in proportion between the two CRS groups. 7 patients have had their leukapheresis product tested for CD8 and CD4. The result showed that the average ratio of CD4⁺ and CD8⁺ T cells (CD4/CD8 T) in the severe CRS group was 1.7 (n=6). Only one patient's CD4/CD8 T was available in the mild group with the value of 0.7 (n=1). The statistical analysis could not be generated owing to the sample size. We have added these data to the revised manuscript (**Line 114~121**).

3. Was the CAR T cell product functionality evaluated? Is that predictive to the subsequent CRS in patients?

Answer: Yes! At the quality control step of manufacturing, the killing capacity of CAR-T cells was evaluated by the apoptotic proportion of BCMA-expressing tumor in co-culture. Phase I and phase II adopted different effector-to-target (E:T) ratios in vitro. In phase I with an E:T ratio of 20:1, the mean apoptotic cell fraction of the mild CRS group was 98% (range: 87% to 100%), and that of the severe group was 95% (range: 83% to 100%), without significant difference. For the 10 investigated patients in phase II, all their CAR-T products were tested for the killing ability in the context of an E:T ratio of 5:1. Only one patient experienced mild CRS, and the rest 9 had CRS graded as level 3 or higher. Though statistical analysis was restricted by the sample size, the average killing capacity of CAR-T cells was 61% and 56% in mild and severe groups, respectively, and wasn't reflective of an evident apoptotic disparity. These data have been added to the revised manuscript (**Line 121~124 and 503~508**).

4. Do the prior reports correlative therapeutic response to CRS? It is not included in

this work on a per patient basis.

Answer: The review's comment is pertinent. We retrieved the therapeutic response to CRS of each patient from our prior reports: PNAS 2019 and JCO 2022. The CRS symptoms, grades, CRS management (Tocilizumab, TNF- α inhibitor, corticosteroid), CRS and disease outcome were specified in a newly added table assigned Supplementary Data 3. And we find no significant difference in MM outcome between the patients with severe (grade >2) and mild CRS (grade ≤ 2), suggesting clinical outcome did not correlate with CRS grade. These data have been added to the revised manuscript (**Line 125~128**).

5. If viral infection is the culprit in the other lethal cases, why is TCR clonality not impacted?

Answer: We thank the reviewer for the important question. We apologize for not elucidating this part clearly. In our study, three cases were reported with viral infection. One (RJ-31) died of uncontrollable hepatic toxicity that resulted from clonal expansion of CAR-T cells in the liver (Fig. 3a, b), whereas the other two (RJ-25 and RJ-27) had transiently mild CAR-T re-expansion without symptomatic CRS and survived with durable responses (Supplementary Fig. 6a~d). With regard to our concern, the appearance of TCR clonality was most likely owing to the *TET2* mutation, which is a genetic event commonly related to clonal hematopoiesis of indeterminate potential (CHIP). CHIP is an indolent stage of hematopoietic disorder and has a 1% risk of progression to leukemia per year. Notably, disruption of the *TET2* gene can not only induce myeloid disease (Jaiswal, S., et al. *N. Engl. J. Med.*, 2014) but also promote CAR-T cell proliferation (Fraietta, J. A., et al. *Nature*, 2018). Our data demonstrated that RJ-31 possessed a somatic *TET2* mutation in the pre-treated peripheral blood with a higher variation fraction in CD3⁺ T cells than the other nucleated cells. Most likely, it constituted a risk of wild outgrowth of CAR-T cells derived from the autologous lymphocytes under certain circumstance, such as viral infection. Nevertheless, the other two patients were not detected with somatic *TET2* mutation, as well as other gene mutations associated with CHIP including *DNMT3A* and *ASXL1*, so the risk of TCR clonality formation was largely minimized. Therefore, the genetic defect was considered the overriding determinant to drive the clonal proliferation of CAR-T, and the viral infection was probably an antigen stimulus to trigger adaptive immunity with CAR-T cells being a part. We have added the above statement to the Discussion section (**Line 404~417**).

6. Are CAR T cells and endogenous T cells reactive to virus in those cases?

Answer: We're grateful for this key question. To evaluate the T cell response to the

virus in patients, the number of CAR-T and endogenous T cells, along with the CD8/CD4 ratio were assessed. In RJ-25 and RJ-27, viral infection occurred at half a year post CAR-T infusion. At the meantime, a CAR-T re-expansion was measurable in both with a dramatic higher CD8⁺ subset level than CD4⁺ counterpart (Supplementary Fig. 6). The endogenous T (CAR-negative T cells) showed the similar tendency compared to the early treatment phase (as shown below, as well as in the **revised Supplementary Fig. 11**). RJ-31 suffered from viral infection on day 20~26, the temporal change in his T cell phenotype better reflected the T cell response to the virus. The data demonstrated an increase in the cell number of both endogenous T and CAR-T cells predominately with the CD8⁺ subset (revised Supplementary Fig. 11, Fig. 3b). Taken together, the three cases shared a common feature, which was, a high CD8 CAR-T and endogenous T cell subpopulation level at virus infection. This phenomenon indicated a competent immunity against the exogenous microbe and the relevant data have been supplemented to the revised manuscript (**Line 319~327**). Owing to the inadequacy of the patients' samples, *in vitro* assays to detect T-cell activation (e.g., CD25, CD69) and killing potential (e.g., granzyme B and perforin) would be taken into account in future study.

revised Supplementary Fig. 11

7. What is TET2 VAF % in CD3-negative populations at later times?

Answer: We thank the reviewer for the key question. On day33, an important later time point, CAR-T in RJ-31 accounted for 47.5% of total T cells. Multiple components of PBMCs on day33 were purified for a deep-resolution detection of

$TET2^{mut}$ VAF by ddPCR. The result showed that $TET2^{mut}$ VAF of CD3-negative subset was 0.4% on day 33, which was basically close to the VAF (0.1%) at pre-treatment. It indicated that $TET2$ mutation was unable to drive non-T hematopoietic cells clonal expansion under the circumstance of microbial infection. We have supplemented this data to the revised Figure 4j, revised Supplementary Fig. 10e and the revised manuscript (Line 346~352).

revised Fig. 4j

revised Supplementary Fig. 10e

Reviewer #4 (Remarks to the Author):

The work proposed by the authors is interesting and allow inspecting the overall landscape of the cytokine profile in those patients reporting one of the most common adverse effect of the cytokine release syndrome after the CAR-T therapy. In the following, I report some crucial points the authors should be considered.

1. The cytokine profile is analyzed without considering the time properly. Specifically, the heatmap of Figure 1C is difficult to interpret since the reader completely lost the temporal evolution of the data. The comments reported are not accurate. For example: on page 6 from line 10 to line 12 it is not completely true since the upregulated molecules do not follow the CRS grade for all the subjects. I strongly suggest using some computational approaches designed to specifically analyze longitudinal data, as well as the framework published by Pernice et al doi: 10.1093/bioinformatics/btad201. Using methodology as CONNECTOR could allow you to stratify the patients according to the cytokine expression and then identify a

relation among cytokines, CRS, and specific groups of patients.

Answer: We are sincerely grateful for the reviewer's constructive comment. The algorithm CONNECTOR you recommended is fairly helpful to objectively identify the relationship between cytokine expression and CRS level.

We first implemented an unsupervised clustering approach to categorize the 61 cytokines into five distinct cytokine clusters (CCs). As indicated in the **revised Figure 1c** and **revised Supplementary Fig. 2b**, CC3, CC4, and CC5 display positive correlations with CRS severity, while CC2 demonstrates a negative correlation ($P < 0.05$).

Of particular note, CC4 and CC5 exhibit the most robust correlations with CRS severity. To interpret the correlation of cytokine expression and CRS level, we adopted the computational methodology CONNECTOR to stratify patients based on their affiliation with CC4 and CC5. Utilizing the mean \log_2FC as the observation, we performed 1,000 iterations using the ClusterAnalysis function within CONNECTOR. This analysis effectively partitioned the 26 CAR-T patients into two groups: Group1 (comprising 8 patients: CZ-02, CZ-03, JS-06, RJ-01, RJ-02, RJ-04, RJ-22, RJ-30) and Group2 (comprising 18 patients: CZ-01, JS-02, JS-03, JS-04, JS-05, JS-07, JS-08, JS-09, RJ-03, RJ-05, RJ-23, RJ-24, RJ-25, RJ-26, RJ-27, RJ-29, RJ-31, RJ-32), as illustrated in the **revised Supplementary Fig. 2c~d**.

Further analysis of the mean \log_2FC of cytokines revealed a distinct cytokine kinetic mode of Group1 from that of Group2, as shown in the **revised Supplementary Fig. 2e~f**. The patients in Group1 displayed a higher peak level of CC3, CC4, and CC5 cytokines, as well as the whole, as compared with Group2. Based on this grouping, we can clearly distinguish the temporal changes in CRS grading during the treatment course. The patients in Group1 experienced severe CRS which peaked around day 12, whereas the patients in Group2 had a relatively mild CRS with a low peak around day 9 (**revised Supplementary Fig. 2g**). It demonstrated that the temporal dynamics of cytokine secretion was highly consistent with CRS grade regardless of individual discrepancy. We have incorporated these important findings into the revised Figure 1 and Supplementary Fig. 2 (**Line 143~161 and 537~546**). We believe that the above data have enhanced the clarity and depth of our analysis. We also would like to extend our sincere gratitude to the reviewer for introducing such a valuable tool, which has greatly enriched our understanding of the temporal evolution of cytokine expression upon CAR-T cells treatment.

revised Figure 1c

revised Supplementary Figure 2b

revised Supplementary Figure 2c-g

2. Moving to the RNASeq analysis. The authors did not specify which is the design formula used in the differential expression analysis of DESeq2. Since in DESeq2 it is necessary to define two groups of comparison, how do the authors define these groups?

Answer: We appreciate the reviewer's advice that helps us to better interpret the workflow of the RNA-seq analysis. In the revised manuscript, a description of the workflow was provided to specify the calculation of the differentially expressed genes (DEGs) and the soft clustering using the mfuzz algorithm:

STEP 1: Raw Expression Data Filtration

We performed a filtration process to exclude genes with low expression, defined as those expressed in less than 30% of the samples. We used the FPKM expression profile for this calculation.

STEP 2: Definition of Comparable Groups and Calculation of DEGs

To ensure comprehensive analysis, we enrolled patients with CRS and conducted longitudinal observations. We categorized the observation period into distinct time intervals: Pre-conditioning, Day0, Day3~5, Day6~9, Day10~12, Day13~15, and Day20~21. For each pair of consecutive observation time points, we calculated DEGs independently. To enhance the reliability of the DEG analysis, we utilized three widely recognized R packages: DESeq2, limma, and edgeR.

As an example, consider the comparison between Day0 and Day3~5. DEGs between these time points were identified based on the following criteria: i) exhibiting a significant difference with adjusted $p < 0.05$, and ii) being identified by three packages. We repeated this process for each pair of observation time points, resulting in the collection of DEGs for each pairwise comparison (21 DEG gene sets among 7 observation time points). A total of 10,917 DEGs were finally identified through combining results from three packages.

STEP 3: Soft Clustering of DEGs

We used the average expression of the 10,917 DEGs at each time point as input for the Mfuzz package. The soft clustering method was employed to investigate the expression patterns of DEGs over time, effectively grouping them based on consistent expression trends.

STEP 4: Identification of Core Genes in Each Gene Cluster

Following the soft clustering, we identified 8 gene clusters. Genes with a α score > 0.6 (indicating higher membership within their clusters) were considered core genes and used for subsequent functional enrichment analysis.

In summary, within the DESeq2 analysis, we have employed cyclic computation to calculate groups, with group definitions based on the observation time. The reviewer's point is immensely valuable in improving the clarity and comprehensibility of our analysis. The revised Method (**Line 562~570 and 578~579**) and Code Availability (**Line 735**) sections have detailed data filtration, formula design, DEG calculation, Mfuzz clustering analysis, and enrichment analysis available on GitHub (https://nrctm-bioinfo.github.io/MM_CART_project/index.html) to ensure transparency and reproducibility of the data assessment, and we hope the explanation

could fully address the reviewer's concerns.

3. Moreover, the authors report that the DE genes were also calculated from the other two packages limma and edgeR, also in this case it is not specified how the design matrix is defined. This is an important point: generally, limma and edgeR are both based on an ANOVA analysis (based on the study of the variance) so why did the authors decide to use two different libraries that used the same approach, aka linear model?

Answer: The reviewer's question is fairly important. We provided an explanation for a clear presentation of the steps involved in DEG calculation as follows:

In our analysis using limma and edgeR, we defined the design matrix as 'model.matrix(~0+Group)', with 'Group' representing two different observation time points, such as Day0 and Day3~5. This approach allowed us to investigate DEGs between any two observation time points. The use of a linear model was motivated by the following reasons:

i) DEGs Calculation Across Different Time Points:

We aimed to thoroughly explore dynamic changes across various observation time points. Therefore, we included DEGs between any two time points for further analysis. The workflow for both limma and edgeR was based on the guidelines provided in the respective software packages.

ii) High-Confidence DEGs Selection:

To ensure the robustness of our results, we employed three well-established and recommended methods (DESeq2, limma, and edgeR) for RNA-seq analysis to identify significant DEGs (adjusted $p < 0.05$). As demonstrated in Supplementary Fig.4a, the majority of DEGs were consistently identified by all three methods, indicating a high level of confidence in our DEG selection process.

We appreciate the reviewer's meticulous attention to the DEG calculation process, and we have incorporated the detailed calculation steps in the revised Methods section (**Line 562~570**) and Code Availability (**Line 735**) to enhance the clarity and transparency of our analysis.

4. For what concern the design matrix, since it is possible to design a higher order time series matrix, which design matrix whose implemented by the authors?

Answer: We sincerely appreciate the reviewer's careful consideration and interest in our DEG calculation design. As explained in our previous answers, we conducted DEG analysis between any two observation time points, including those at higher-order time intervals. To delve deeper into the dynamic changes within our longitudinal data, particularly at higher-order time points, we employed Mfuzz, a well-established unsupervised classification methodology.

We are grateful to the reviewer for the insightful comments regarding the DEG calculation design, as those have significantly guided us to enhance the clarity and

depth of our analysis of dynamic changes in RNA-seq during CRS development.

5. At the end of the definition of DE genes from all three packages, how do the authors select the genes for the GSEA analysis? The intersection of the DE genes obtained from the three methods?

Answer: We extend our gratitude to the reviewer for raising this critical point. Meanwhile, we apologize for neglecting to clarify the DEG definition and GSEA analysis. The explanations were provided as below:

DEG Definition and Soft Clustering:

The DEGs identified by DESeq2, limma, and edgeR were specifically utilized for soft clustering in Mfuzz analysis. We employed this approach to explore the dynamic expression patterns of these DEGs across the different observation time points, providing valuable insights into the temporal dynamics of gene expression.

GSEA Analysis:

The GSEA analysis was conducted independently, and the genes used for this analysis were selected from the FPKM profile. Genes with low expression (expressed in less than 30% of the samples) were excluded. During the GSEA analysis at various observation time points, genes were pre-ranked based on their \log_2FC , which was calculated by comparing their expression levels with the pre-conditioning time point.

To assess the statistical significance of enrichment, we performed 1,000 permutations and subsequently adjusted the enrichment P-values using the False Discovery Rate (FDR) method. Pathway direction is the median \log_2FC of significant transcripts (core enrichment genes) relative to the baseline in each pathway. In this visualization, an adjusted $p < 0.05$ with pathway direction > 0 indicated activation, while an adjusted $p < 0.05$ with pathway direction < 0 indicated repression (as demonstrated in Figure 3d). These details have been added to the revised methods section (**Line 562~570, 578~579 and 586~593**).

6. Does the analysis of the time series data by soft clustering method consider all genes or only the DE genes?

Answer: We thank the reviewer for the comment regarding the gene selection for soft clustering. In our analysis, only the DEGs, as identified by DESeq2, limma, and edgeR, were deployed for the subsequent soft clustering analysis. Indeed, this approach has allowed us to focus on the dynamic expression patterns of genes that exhibited significant changes during CAR-T therapy and CRS development. We have carefully revised our manuscript accordingly (**Line 562~570 and 578~579**).

REVIEWERS' COMMENTS

Reviewer #2 (Remarks to the Author):

no comments

Reviewer #3 (Remarks to the Author):

Thank you for the comprehensive revision of this manuscript. A single additional point should be addressed given the revision experiments:

1. Given the lymphocyte count differences at pre-treatment, it may be useful to provide also the absolute counts of CD8 and CD8, not just ratios.

Reviewer #4 (Remarks to the Author):

I really appreciated all the additional analysis that the authors did. They worked in a very professional and accurate way and they addressed all my questions and comments.

Reviewer #3 (Remarks to the Author):

Thank you for the comprehensive revision of this manuscript. A single additional point should be addressed given the revision experiments:

1. Given the lymphocyte count differences at pre-treatment, it may be useful to provide also the absolute counts of CD8 and CD4, not just ratios.

Answer: The reviewer's comment is constructive. There were 16 patients having the available data of CD8 and CD4 frequencies in CAR-T products. The result showed that the mean absolute counts of CD8⁺ and CD4⁺ CAR-T cells of the infused products in the mild CRS group were respectively $0.46 \times 10^6/\text{kg}$ and $0.25 \times 10^6/\text{kg}$, and those in the severe group were respectively $0.43 \times 10^6/\text{kg}$ and $0.32 \times 10^6/\text{kg}$. Neither CD8⁺ nor CD4⁺ reached significant difference in absolute counts between the two CRS groups. We have added this result to the revised manuscript (**Line 128-132**). Meanwhile, 7 patients have had their leukapheresis product tested for CD8 and CD4 expression. The result showed that the average frequencies of CD4⁺ and CD8⁺ T cells in the severe CRS group was $1.76 \times 10^8/\text{L}$ and $1.72 \times 10^8/\text{L}$ (n=6). Only one patient's CD4⁺ and CD8⁺ T cells were available in the mild group with the absolute counts of $3.13 \times 10^8/\text{L}$ and $4.61 \times 10^8/\text{L}$ (n=1). The statistical analysis could not be generated owing to the sample size.